**Planktonic foraminifera assemblage composition and flux dynamics inferred from an annual**
**sediment trap record in the Central Mediterranean Sea**
Thibauld M. Béjard[1*], Andrés S. Rigual-Hernández[1], Javier Pérez Tarruella[1], José-Abel Flores[1], Anna
Sanchez-Vidal[2], Irene Llamas-Cano[2], Francisco J. Sierro[1]
1. Área de Paleontología, Departamento de Geología, Universidad de Salamanca, Salamanca, Spain
2. GRC Geociències Marines, Departament de Dinàmica de la Terra i de l'Oceà, Universitat de
Barcelona, Spain
**\* Correspondence:** Thibauld M. Béjard (thibauld.bejard@usal.es)
**Keywords:** sediment trap - Sicily Channel - Mediterranean Sea - planktonic foraminifera - seasonal
variations - environmental change
**Abstract**
The Sicily Channel, located in the Central Mediterranean Sea, represents a key point for the regional
oceanographic circulation as it is considered the sill that separates the western and eastern basins.
Therefore, it is considered a unique zone regarding the well-documented west-to-east
Mediterranean productivity gradient. Here we present a time series of settling planktonic
foraminifera assemblages from November 2013 to October 2014. 19 samples from the sediment
trap C01 deployed at a water depth of around 400 m have been used. More than 3700 individuals
and 15 different species have been identified. *Globorotalia inflata*, *Globorotalia truncatulinoides*,
*Globigerina bulloides*, *Globigerinoides ruber* and *ruber* (pink) were the five main species identified,
accounting for more than 85% of the total foraminifera.
The total planktonic foraminifera flux mean value was 630 shells $m^{-2}$ $d^{-1}$, with a minimum value of
45 shells $m^{-2}$ $d^{-1}$ displayed during late autumn 2013 and a maximum of 1890 shells $m^{-2}$ $d^{-1}$ reached
during spring 2014. This is likely due to the regional oceanographic configuration and the marked
seasonality in the surface circulation. During spring and winter, the Atlantic waters dominate the
surface circulation, bringing cool and nutrient enriched waters. This results in a planktonic
foraminifera flux increase and a dominance of western basin taxa. During summer and autumn, the
circulation is dominated by the eastern warm and oligotrophic Levantine water, which leads to a
planktonic foraminifera flux decrease and the dominance of easter basin species. Our comparison
with satellite derived SST and chlorophyll-*a* data showed that *G. inflata* was associated with cool
and nutrient rich conditions, while both *G. ruber* morphotypes were associated with warm and
oligotrophic conditions. However, no trends were identified for *G. truncatulinoides* or *G. bulloides*.
As the latter species flux increased coincidently with that of benthic foraminifera one, we considered
that this species might have a resuspended origin.
The comparison of the Sicily Channel data with other Mediterranean time series indicates that the
annualized planktonic foraminifera flux was lower than in the westernmost Alboran Sea but higher
than in the easternmost Levantine basin. The Sicily Channel species diversity was the highest among
the compared zones, highlighting the influence of the different basins and its transitional aspect
from a planktonic foraminifera population perspective.
Finally, we compared the settling planktonic foraminifera assemblage with the assemblages from
seabed sediment located in the vicinity of the Sicily Channel. The differences with the seabed
populations varied according to the sites studied. The deep-dwelling species dominated the settling
assemblages samples, while eutrophic and oligotrophic species were more abundant in the
sediment. Finally, a high-resolution chronology comparison allowed to show that this planktonic
foraminifera population shift likely developed during the late Holocene prior to the industrial period,
however, its causes remain uncertain.
**1. Introduction**
Planktonic foraminifera are a group of marine calcareous single-celled protozoans with a
cosmopolitan distribution. Around 50 morphospecies of planktonic foraminifera have been
described in today's oceans (Schiebel and Hemleben, 2017), and although most of those species are
surface dwellers, some species can be found in waters below 2000 m (Schiebel and Hemleben,
2005). Their abundance and distribution are affected by a wide array of factors, such as
temperature, salinity, chlorophyll-$a$ and nutrient concentrations, among others (Hemleben et al.,
1989; Schiebel and Hemleben, 2005). According to Schiebel, (2002), the production and export of
their calcareous shells account for 23 to 56% of the open marine $CaCO_3$ flux, thereby playing a key
role in the marine carbon cycle. Moreover, the high preservation potential of their shells makes
them one of the most used groups for multi-proxy studies. Numerous paleoclimatic (e.g. Barker and
Elderfield, 2002; Lirer et al., 2014; Margaritelli et al., 2020; Sierro et al., 2005) and
paleoceanographic (Cisneros et al., 2016; Ducassou et al., 2018; Margaritelli et al., 2022; Toucanne
et al., 2007) reconstructions have used planktonic foraminifera as a proxy. In addition, their capacity
to reflect the water column's chemical properties has propelled studies that have focused on the
impact of recent climate and environmental variability on the water column in different parts of the
ocean (e.g. Azibeiro et al., 2023; Beer et al., 2010; Bijma et al., 2002; Chapman, 2010; Marshall et
al., 2013; Osborne et al., 2016). As marine calcifying organisms, they are considered particularly
vulnerable to the ongoing ocean warming and acidification (Bijma et al., 2002; Fox et al., 2020). Shell
calcification of several foraminifera species has been showed to decrease in response to ocean
acidification, and therefore, changes in the weight of their shells are considered an indicator of the
ocean acidification impact on different timescales (Béjard et al., 2023; de Moel et al., 2009; Fox et
al., 2020; Kroeker et al., 2013; Moy et al., 2009; Pallacks et al., 2023). In contrast, ocean warming
has been proposed to produce an opposite effect on foraminifera calcification, as some studies have
documented that an increase in water temperature results in larger shells and enhanced growth
rates (Lombard et al., 2011, 2009; Schmidt et al., 2006).
Despite the wide array of studies focused on planktonic foraminifera ecology and distribution,
several aspects of their ecology remain uncertain, such as their ecological tolerance limits (Mallo et
al., 2017), their geographical and temporal distributions and contribution to the marine
biogeochemical cycles (Jonkers and Kučera, 2015). As major contributors to the pelagic calcite
production (Schiebel, 2002), understanding their life cycle on different time scales is essential for

constraining the role they play in the marine carbon cycle and the impact of environmental change on these organisms. In this regard, sediment traps represent a powerful tool to improve our knowledge of planktonic foraminifera ecology and their impact on the biogeochemical cycles, as they allow the monitoring of foraminifera shell fluxes for extended periods, thereby allowing to document their seasonal and interannual variability and estimate their contribution to annual budgets of carbonate export to the seafloor (Jonkers et al., 2019).

The Mediterranean Sea is a semi-enclosed sea often considered a "miniature ocean" (Bethoux et al., 1999) from an oceanographic point of view or a "laboratory basin" (Bergamasco and Malanotte-Rizzoli, 2010) for studying processes occurring on a global scale. In addition, it is supersaturated regarding calcite (Álvarez et al., 2014), a key aspect in foraminifera studies, as this parameter favors shell preservation and represents one of the main environmental controls on planktonic foraminifera abundance and calcification (Aldridge et al., 2012; Marshall et al., 2013; Osborne et al., 2016). These features make it an interesting zone of the global ocean to study the life cycle and seasonal response to changing environmental conditions of calcifying plankton. The Sicily Channel, in the central Mediterranean, is the sill that divides the Mediterranean into its western and eastern basins. It is a choke point for the regional surface and deep-water circulation (Malanotte-Rizzoli et al., 2014; Pinardi et al., 2015) and a transition region regarding the well-known west-to-east oligotrophy gradient, functioning as a "biological corridor" (Siokou-Frangou et al., 2010) known in the Mediterranean (Navarro et al., 2017).

Despite these characteristics, time series that focused on planktonic foraminifera in the Mediterranean Sea are scarce. So far, the best monitored regions are the Alboran Sea (Bárcena et al., 2004; Hernández-Almeida et al., 2011), the Gulf of Lions (Rigual-Hernández et al., 2012), and more recently, the Levantine Basin (Avnaim-Katav et al., 2020). The latter studies showed that planktonic foraminifera followed a unimodal distribution with maximum shell export occurring during the months of April-May, February-March and February respectively, which agreed with the local hydrographic conditions. However, the central Mediterranean remains understudied and poorly documented regarding both continuous time series and planktonic foraminifera dynamics.

Therefore, this work aims to provide new planktonic foraminifera data from a sediment trap mooring line located in the Channel of Sicily to improve the current knowledge about their community composition and seasonal patterns in the central Mediterranean. For that purpose, here we document the magnitude and composition of planktonic foraminifera fluxes identified in the >150 μm fraction (i.e. the most commonly used size fraction for studying planktonic foraminifera distribution) from November 2013 to October 2014. We compare our planktonic foraminifera data with a suite of environmental parameters to assess the main environmental drivers that control the seasonal variations in the composition and abundance of the sinking planktonic foraminifera assemblages. To provide further insight on a regional and global scale of the planktonic foraminifera association and fluxes identified here, we compare our data with other time series from the Mediterranean, Atlantic Ocean and other regions of the world's oceans. Lastly, we compared the assemblages collected by the sediment with seabed sediment located in the vicinity of the Sicily Channel to document the potential shift in recent planktonic foraminifera populations.

## 2. Study area

The Mediterranean is an elongated, semi-enclosed sea with an anti-estuarine circulation. It is considered to be a concentration basin (Bethoux et al., 1999) in which the evaporation exceeds the freshwater inputs, forcing a negative hydrological balance (Robinson and Golnaraghi, 1994). This negative balance is compensated by the entrance of surface oceanic water from the Atlantic Ocean through the Channel of Gibraltar. The colder and nutrient richer Atlantic Waters (AW) spread eastward into the Mediterranean basin (Millot, 1991; Pinardi et al., 2015), where they progressively become warmer, saltier and more oligotrophic as they mix with resident waters (Modified Atlantic Waters – MAW. Also known as Atlantic Waters – AW). MAW circulate following a cyclonic circuit along the Algerian coast (Algerian Current – AC) (Malanotte-Rizzoli et al., 2014; Millot, 1999) and divide into two main branches at the entrance of the Sicily Channel (Figure 1a). One of these branches spreads into the northwestern part of the basin, into the Tyrrhenian Sea, where it continues its path cyclonically. The second branch flows south of Sicily into the Ionian Sea (Lermusiaux and Robinson, 2001). In the Sicily channel itself, the water masses are split again in two different streams (Béranger et al., 2004): (i) the Atlantic Tunisian Current (ATC) that flows to the southeast following the African coast; and (ii) the Atlantic Ionian Stream (AIS) that flows into the deep eastern part of the basin (Figure 1b) and contributes to the MAW transport in the eastern Mediterranean (Jouini et al., 2016; Lermusiaux and Robinson, 2001).

The Sicily Channel is located in the central Mediterranean (Figure 1a) and acts as a sill that topographically separates the western and eastern Mediterranean basins. The circulation through the Sicily Channel is characterized by water masses that flow in opposite directions at different depths of the water column (Béranger et al., 2004; Garcia-Solsona et al., 2020; Pinardi et al., 2015; Schroeder et al., 2017). The Levantine Intermediate Water (LIW), which enters the Channel from the Ionian Sea, occupies the deeper part of the water column along with occasional thin Eastern Mediterranean Deep Water layers (Gasparini et al., 2005; Lermusiaux and Robinson, 2001). The Ionian Water (IW) can be present at intermediate depths (Figure 1), while the MAW cover the surface to subsurface part of the water column (Garcia-Solsona et al., 2020; Warn-Varnas et al., 1999). Temperature and salinity range from 15-17 °C and 37.2-37.8 psu for the MAW, 15-16.5 °C and 37.8-38.4 psu for the IW and 13.7-13.9°C and 38.7-38.8 psu for the LIW (Astraldi et al., 2002; Bouzinac et al., 1999; Robinson et al., 1999). Lastly, it is important to note, that the surface circulation in the Sicily Channel presents a large seasonal variability concerning the water masses distribution (Béranger et al., 2004; Lermusiaux and Robinson, 2001). Surface circulation experiences a substantial seasonality in the Sicily Channel: during late autumn to late spring, the MAW dominate the surface circulation, allowing nutrient and chlorophyll-enriched waters to enter the Channel (Astraldi et al., 2002; D'Ortenzio, 2009). In turn, summer and autumn are dominated by LIW waters. Deep-water circulation remains relatively stable on a seasonal scale (Béranger et al., 2004) with a continuous LIW presence over the year. Finally, during summer, an upwelling settles in the Sicily Channel, allowing the impoverished LIW to reach the surface (Lermusiaux and Robinson, 2001).

Regarding its nutrient distributions, the Mediterranean Sea is generally considered an oligotrophic to ultraoligotrophic sea (Krom et al., 1991). However, this oligotrophy is not homogenous and displays a clear west-to-east gradient which is reflected in the nutrient concentration and algal

biomass accumulation derived from colour remote sensing (Navarro et al., 2017; Siokou-Frangou et
al., 2010). The eastern part of the Mediterranean is considered to be more nutrient depleted than
the western part of the basin (Krom et al., 2005; Raimbault et al., 1999), with N:P ratios around 50:1
(Krom et al., 2005). At times of maximum annual algal concentration, primary productivity (PP) in
the Levantine Basin reaches values of ca. 0.1 g C m$^{-2}$d$^{-1}$ (Hazan et al., 2018). This value is substantially
lower than those recorded in the high productivity regions of the western basin such as the Gulf of
Lions, ca. 0.4-0.65 g C m$^{-2}$d$^{-1}$ (Gaudy et al., 2003; Rigual-Hernández et al., 2012), or the Alboran Sea,
ca. 0.3-1.3 g C m$^{-2}$d$^{-1}$ (Bárcena et al., 2004; Morán and Estrada, 2001) during the corresponding
period.

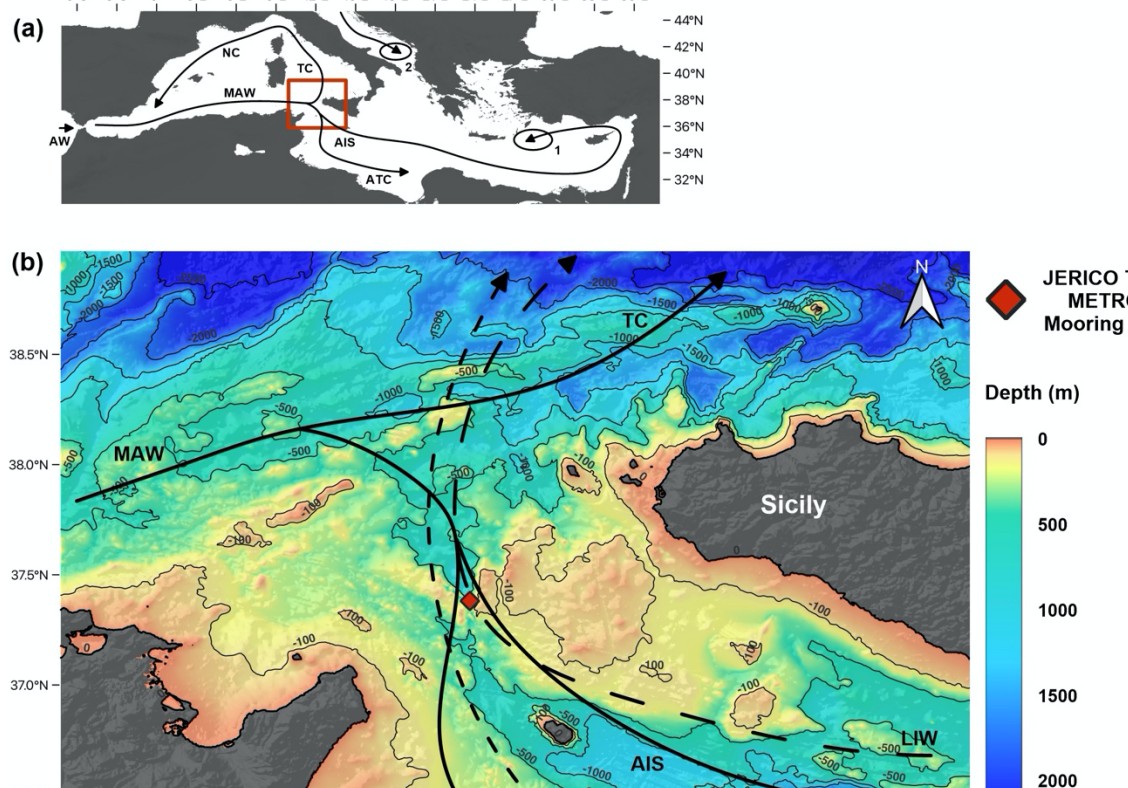

**Figure 1. (a)** Mediterranean Sea general surface circulation (Astraldi et al., 2002; Béranger et al., 2004;
Incarbona et al., 2011; Macias et al., 2019) and location of the study zone. The ellipses show the deep-
water formation zones for the LIW (1) and the EMDW (2). **(b)** Regional oceanographic and geographic
setting of the Sicily Channel. The red diamond represents the location of the JERICO TNA METRO C01
mooring line. Black continuous lines represent the surface circulation dominated by the Atlantic Ionian
Stream (AIS) and the Atlantic Tunisian Current (ATC); while dashed lines show deep-water circulation
influences by the Levantine Intermediate Water (LIW) and the Eastern Mediterranean Deep Water
(EMDW). The difference in the dashed lines period stands for the occasional aspect of the EMDW. The
topographic model was downloaded from the GEBCO database.
## 3. Material and methods



### 3.1. Field experiments



The sediment trap (Figure 1) was deployed in the C01 mooring line maintained by ISMAR-CNR in the
Sicily Channel (37.38°N, 11.59°E) thanks to a TransNational Access (TNA) call in the FP7 JERICO
project (Mediterranean sediment Trap Observatory). The mooring line was equipped with a
sequential sampling sediment trap located 413 m below the sea surface in a water column of around
450 m deep. The sediment trap was a PPS3/3 model, conical in shape with a 2.5 height/diameter
ratio and equipped with 12 sampling cups. Further information about this sediment trap
configuration and model can be found in Heussner et al., (2006, 1990).
Here we present data from November 2013 to mid-October 2014. The sampling period was 15 to
16 days from November 2013 to July 2014 and from September 2014 to October 2014. Between
July 2014 and September 2014, the sampling was set to 31 days. Before deployment and to limit the
degradation of the material caught, sediment trap sampling cups from both mooring lines were
filled with a 5% formalin solution prepared with 40% formaldehyde mixed with 0.45 $\mu$m filtered
seawater. The solution was then buffered with sodium borate to keep the pH stable and prevent
the dissolution of carbonate.

### 3.2 Processing of sediment trap samples



After the recovery, the cups were stored at 2-4°C until their processing according to the procedure
of Heussner et al., (1990). In the laboratory, the largest swimmers that entered the trap were
removed by wet sieving through a 1 mm nylon and samples were subsequently split into 6 aliquots
using a peristaltic pump. One sub-sample was used for total mass flux measurements, after having
<1mm swimmers and formaldehyde removed.
Another subsample of a total of 19 samples from the sediment trap were processed for
micropaleontological analyses in the micropaleontology laboratory of the Geology department at
the University of Salamanca. The samples consisted of aliquots of 1/6 of the original mooring line
cups and were preserved in seawater, with a pH between 7.6 and 7.8. All samples were first wet
sieved to separate the <63$\mu$m fraction and then dry sieved to separate the 63-150 and >150 $\mu$m
fractions. The washing was carried out with a potassium phosphate-buffered solution (pH= 7.5) to
prevent carbonate dissolution.

### 3.3 Planktonic foraminifera identification, flux calculations and imaging



The planktonic foraminifera identification (Plate 1) and counting to the species level were carried
out in the >150 $\mu$m fraction using a microscope (Leica Wild M3B). To have a representative picture
of the planktonic foraminifera population, the complete samples were analyzed (i.e. no splits were
applied). Identification was carried out according to Schiebel and Hemleben, (2017). A total of 15

species were identified (Plate 1): *Globigerinella siphonifera*, *G. calida*, *Globigerinoides sacculifer*, *G. ruber*, *G. ruber* (pink), *Globoturborotalita tenella*, *G. rubescens*, *Orbulina universa*, *Globorotalia truncatulinoides*, *G. inflata*, *G. scitula*, *Globigerina bulloides*, *G. falconensis*, *Neogloboquadrina incompta* and *Turborotalita quinqueloba* (Plate 1). In addition, benthic foraminifera shells were identified to the lowest taxonomic level possible and counted. The 150 μm size limit was used to compare our results with other time series and seabed sediment populations as it is widely used in planktonic foraminifera studies, however, we acknowledge that some "small-sized" species such as *N. incompta* and *G. tenella* may be undersampled as their adult size tends to be smaller (Chernihovsky et al., 2023).

The foraminifera fluxes were calculated according to the following formula:

$$PF \ (shells \ m^{-2} \ d^{-1}) = \frac{(N \ x \ aliq.) \ x \ SD^{-1}}{0.1256}$$

"PF" stands for planktonic foraminifera, "N" accounts for the number of individuals identified, "aliq." refers to the aliquot (1/6 for all samples) and "SD" represents the sampling interval that the sediment trap cup stayed open. Relative abundance for each species was also calculated for all samples.

Here we refer to the planktonic foraminifera collected by the sediment trap as the settling assemblage.

Lastly, to describe the seasonal flux variations and to put our results into a regional context and be coherent with previous studies, each season was defined as spring (March–May), summer (June–August), autumn (September–November) and winter (December–February).

To showcase the species collected by the traps (Plate 1), foraminifera imaging was carried out using a Nikon SMZ18 stereomicroscope equipped with a Nikon DS-Fi3 camera and the image processing software NISElements (version 5.11.03).

**3.4. Satellite-derived environmental parameters**

To assess the possible relationship of planktonic foraminifera fluxes with environmental variability, satellite-derived chlorophyll-*a* and Sea Surface Temperatures (SSTs) were retrieved from global data sets. Satellite-derived chlorophyll-*a* concentration (mg m$^{-3}$) was obtained from MODIS L3m satellite through NASA's Giovanni web interface with an 8-day and 4 km resolution for a 0.2 x 0.2° area around the mooring location between 01/10/2013 to 01/11/2014. Additionally, sea surface temperature SST (°C) were also obtained from the same site with the same resolution to use as a proxy for water temperature and water column stratification.

**3.5 Planktonic foraminifera flux and surface sediment data from other Mediterranean settings**

In order to put into context our observations with the regional variability of planktonic foraminifera communities in the Mediterranean Sea, modern planktonic foraminifera flux datasets were retrieved from different sites. Foraminifera fluxes of: (i) the Levantine basin (LevBas) were obtained from Avnaim-Katav et al., (2020); (ii) the Gulf of Lions (stations Planier - PLA, and Lacaze Duthiers -

LCD) from Rigual-Hernández et al., (2012); (iii) and the Alboran Sea (stations ALB 1F and ALB 5F)
from both Bárcena et al., (2004) and Hernández-Almeida et al., (2011). The foraminifera fluxes of
the Gulf of Lions and Alboran Sea concerned the >150 μm fraction, while the ones from the
Levantine basin represented the >125 μm fraction (Figure 7).
Core-top data from the ForCenS database (Siccha and Kucera, 2017) was used to compare the
planktonic foraminifera abundance patterns from the C01 mooring line with the seabed sediment.
Only seabed sediment located on a 2.5 degree difference in both latitude and longitude was selected
to compare our data with sites in the vicinity of the Sicily Channel. This corresponded to a total of
16 core-tops part of the MARGO database. The complete details of the latter can be found in the
Supplementary data.
Additionally, the planktonic foraminifera population data from two box-cores analyzed by Incarbona
et al., (2019) were also included: sites 342 (36.42°N, 13.55°E) and 407 (36.23°N, 14.27°E). These two
sites are located in the Sicily Channel and they provide a robust chronology ($^{210}$Pb) that allowed to
document abundance changes across the recent Holocene. The dating covered the years 1558 to
1994 CE. Here we compared the sediment trap from the C01 mooring line samples with the mean
relative abundance from the 23 (site 342) and 24 (site 407) samples available.
Finally, to have a more complete picture of the modern planktonic foraminifera communities
currently living the surface ocean, the annual integrated data of our sediment trap was compared
with the BONGO nets data from Mallo et al., (2017), specifically, with the sample retrieved in the
axis of the Sicily Channel (37.08°N, 13.18°E) in Spring 2013.

**3.6 Statistical analysis**

To have uninterrupted monthly and daily values from NASA's Giovanni environmental parameters
that coincide with the mean sampling date from the sediment trap, a daily resampling has been
carried out using QAnalySeries software.
Pearson correlation and *p*-value tests between the foraminifera abundances and the environmental
parameters (SST and chlorophyll-*a*) were carried out with the Past4 program. A $p < 0.05$ was used
to denote statistical significance.
In addition, a canonical correspondence analysis (CCA) was also used to evaluate the influence of
both SST and chlorophyll-*a* on foraminifera species fluxes. A CCA is a correspondence analysis of a
species matrix where each site has given values for one or more environmental variables (SST and
chlorophyll-*a* concentration in this case). The ordination axes are linear combinations of the
environmental variables. A CCA is considered an example of direct gradient analysis, where the
gradient in environmental variables is known and the species abundances/fluxes are considered to
be a response or to be affected by this gradient (Nielsen, 2000).
Additionally, to evaluate the magnitude of the foraminifera fluxes across major regions of the
Mediterranean, an estimation of the annual planktonic foraminifera flux (shells m$^{-2}$ y$^{-1}$) was
calculated using the sediment trap data from the literature review and our study. To that purpose,
the data was annualized according to the following formula:
$$Annual\ PFF = \sum(PF\ x\ SD + cPF\ x\ mSD)$$
Where "PFF" stands for planktonic foraminifera flux (shells $m^{-2} d^{-1}$), "SD" accounts for sampling days,
"cPF" represents calculated planktonic foraminifera flux (shells $m^{-2} d^{-1}$) and "mSD" stands for missing
sampling days. "cPF" calculation depended on the site. For the datasets retrieved from the Sicily
Channel and the Levantine basin, less than 20 sampling days were missing, so the corresponding
planktonic foraminifera fluxes were replaced by the mean of the first and last flux values recorded.
The two datasets from the Alboran Sea displayed more than 70 missing days, so the corresponding
flux values used were a mean of the two closest months to the missing data. Concerning the two
time series from the Gulf of Lions, they covered more than one year. Therefore, a mean year was
estimated: a mean monthly flux value was calculated for all 12 months based on all the available
measurements and then multiplied by the corresponding mean duration of each month, and then,
all monthly fluxes were added together.
To compare the species richness and diversity across the previously described sites, Simpson (D) and
Shannon/Weiner (H/W) indexes were calculated. Here, we reported the inverse Simpson index (1-
D). None of these indexes were calculated for the Alboran Sea sites (ALB 1F and ALB 5F) because
only information about the four main species was documented (Bárcena et al., 2004; Hernández-
Almeida et al., 2011).
Finally, the squared chord distance (SCD) between the C01 sediment trap and every core top sample
downloaded from the ForCenS database (Siccha and Kucera, 2017) planktonic foraminifera relative
abundance was calculated. It is a widely used metric in palaeoecological and paleontological studies
as it is the most effective index for identifying the closest analogues in planktonic foraminifera
datasets (Prell, 1985). This is mainly because it shows the best balance in weighing the contribution
of abundant and rare species in a given association (Jonkers et al., 2019). In this study, SCD values
lower than 0.25 have been considered as reliable analogues (Ortiz and Mix, 1997).


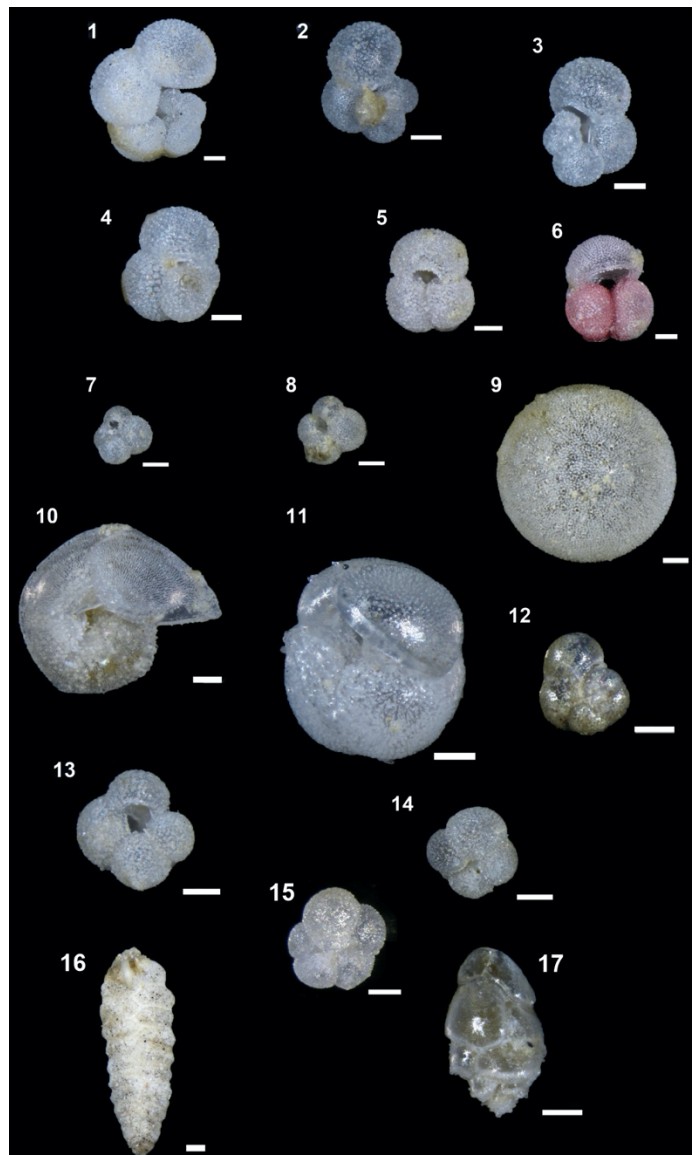

**Plate 1.** Planktonic **(1-15)** and the most common benthic foraminifera **(16-17)** species trapped in the sediment trap in mooring line C01. The white scale bars on all figures represent 100 μm. **(1)** *G. siphonifera,* side view. **(2)** *G. calida*, umbilical view. **(3)** *G. calida*, apertural view. (**4**) *G. sacculifer*, umbilical view. (**5**) *G. ruber*, umbilical view. **(6)** *G. ruber* (pink), umbilical view. **(7)** *G. tenella*, umbilical view. **(8)** *G. rubescens*, umbilical view. **(9)** *O. universa*. **(10)** *G. truncatulinoides*, umbilical view. **(11)**. *G. inflata*, apertural view. **(12)** *G. scitula*, umbilical view. **(13)** *G. bulloides*, umbilical view. **(14)** *N. incompta*, umbilical view. **(15).** *T. quinqueloba*, umbilical view. **(16)** *Textularia spp.* **(17)** *Bulimina marginata*, apertural view.

## 4. Results

### 4.1 General considerations of the planktonic foraminifera assemblages

**Table 1.** Counts and key statistics of the planktonic foraminifera species and the benthic foraminifera group from the > 150 μm fraction identified in the 19 sediment trap cups of the C01 mooring line. Mean, maximum (Max), minimum (Min), standard deviation (SD) of the relative abundance and fluxes. Raw counts also include a total and % of the total description. Note that *G. falconensis* was documented but not included in the table due to its scarcity (only one individual was identified).

| | G. sipho. | G. cal. | G. sacc. | G. rub. | G. rub.(p.) | G. ten. | G. rubesc. | O. univ. | G. truncat. | G. inf. | G. sci. | G. bull. | N. inc. | T. quin. | Benthics | Total planktonic |
|---|---|---|---|---|---|---|---|---|---|---|---|---|---|---|---|---|
| **COUNTS (N)** | | | | | | | | | | | | | | | | |
| Mean | 2.5 | 3.1 | 4.1 | 6.5 | 5.2 | 1.1 | 3.7 | 3.9 | 37.0 | 109.2 | 1.3 | 16.2 | 1.5 | 0.5 | 7.4 | 195.9 |
| Max | 6 | 11 | 10 | 22 | 40 | 5 | 9 | 15 | 118 | 456 | 7 | 111 | 8 | 3 | 42 | 633 |
| Min | 0 | 0 | 0 | 1 | 0 | 0 | 0 | 0 | 1 | 1 | 0 | 0 | 0 | 0 | 1 | 14 |
| SD | 1.8 | 2.8 | 3.2 | 5.6 | 9.2 | 1.5 | 2.5 | 4.1 | 33.2 | 132.5 | 2.3 | 26.4 | 2.3 | 1.1 | 9.2 | |
| Total | 48 | 59 | 78 | 124 | 99 | 21 | 71 | 74 | 703 | 2075 | 24 | 307 | 29 | 10 | 141 | 3723 |
| % of total | 1.3 | 1.6 | 2.1 | 3.3 | 2.7 | 0.6 | 1.9 | 2.0 | 18.9 | 55.7 | 0.6 | 8.2 | 0.8 | 0.3 | 3.3 | |
| **ABUNDANCES (%)** | | | | | | | | | | | | | | | | |
| Mean | 2.0 | 2.7 | 2.8 | 5.5 | 5.7 | 0.9 | 4.0 | 3.0 | 20.5 | 41.6 | 1.9 | 7.3 | 1.8 | 0.2 | 5.2 | |
| Max | 7.4 | 10.2 | 8.1 | 16.0 | 32.5 | 8.5 | 14.3 | 16.9 | 46.1 | 72.0 | 8.8 | 26.7 | 21.4 | 1.7 | 12.5 | |
| Min | 0.0 | 0.0 | 0.0 | 0.5 | 0.0 | 0.0 | 0.0 | 0.0 | 7.1 | 1.6 | 0.0 | 0.0 | 0.0 | 0.0 | 0.6 | |
| SD | 2.0 | 2.7 | 2.4 | 4.7 | 10.1 | 1.9 | 4.3 | 3.9 | 9.0 | 24.0 | 3.2 | 6.5 | 4.8 | 0.4 | 3.9 | |
| **FLUXES (shells m$^{-2}$ d$^{-1}$)** | | | | | | | | | | | | | | | | |
| Mean | 7.9 | 10.2 | 13.2 | 19.6 | 15.8 | 3.6 | 12.0 | 11.0 | 113.8 | 354.9 | 3.3 | 57.2 | 5.3 | 1.8 | 24.8 | 629.8 |
| Max | 26.1 | 47.8 | 34.7 | 65.7 | 127.4 | 21.7 | 28.7 | 35.0 | 368.5 | 1361.5 | 22.3 | 482.0 | 34.7 | 13.0 | 182.4 | 1889.9 |
| Min | 0.0 | 0.0 | 0.0 | 3.2 | 0.0 | 0.0 | 0.0 | 0.0 | 3.2 | 3.2 | 0.0 | 0.0 | 0.0 | 0.0 | 3.0 | 44.6 |
| SD | 6.5 | 11.1 | 11.3 | 17.7 | 29.6 | 5.8 | 8.6 | 10.7 | 107.2 | 426.4 | 6.3 | 110.7 | 8.8 | 3.9 | 39.9 | |

A total of 3723 planktonic foraminifera shells and 141 benthic foraminifera were counted. Planktonic foraminifera were identified at the species level, resulting in a total of 15 different species identified (Plate 1). A mean of 196 planktonic foraminifera specimens per sample were identified, with a minimum of 14 individuals in November 2013 and a maximum of 633 individuals in mid-March 2014 (Table 1).

According to the raw counts results, the most abundant species was *G. inflata*, which represented 55.7% of the total planktonic foraminifera individuals. The second most represented species was *G. truncatulinoides*, with 18.9%, followed by *G. bulloides* with 8.2%. These three species alone accounted for more than 80% of the planktonic foraminifera identified. The remaining species abundances were below 5%. *G. ruber*, *G. ruber* (pink), *O. universa*, *G. rubescens* and *G. sacculifer* represented between 2 and 3.3 % of the total individuals. Species like *G. tenella*, *G. scitula*, *N. incompta* and *T. quinqueloba* were very scarce and accounted individually for less than 1% of the total planktonic individuals (Table 1). Finally, only one individual of *G. falconensis* has been identified. Note that *G. inflata*, *G. truncatulinoides* and *G. ruber* were the only species present in all samples. Concerning the differentiation between lobulated and sac-type *Globigerinoides*, we mainly found individuals belonging to the first group, the sac-type individuals were scarce. The latter were identified mainly during summer and autumn.

Finally, the benthic foraminifera only represented 3.3% of the total foraminifera identified and 80%
of the individuals were identified in the two samples retrieved during April 2014 (see Supplementary
data).

**4.2 Total mass and planktonic foraminifera fluxes**

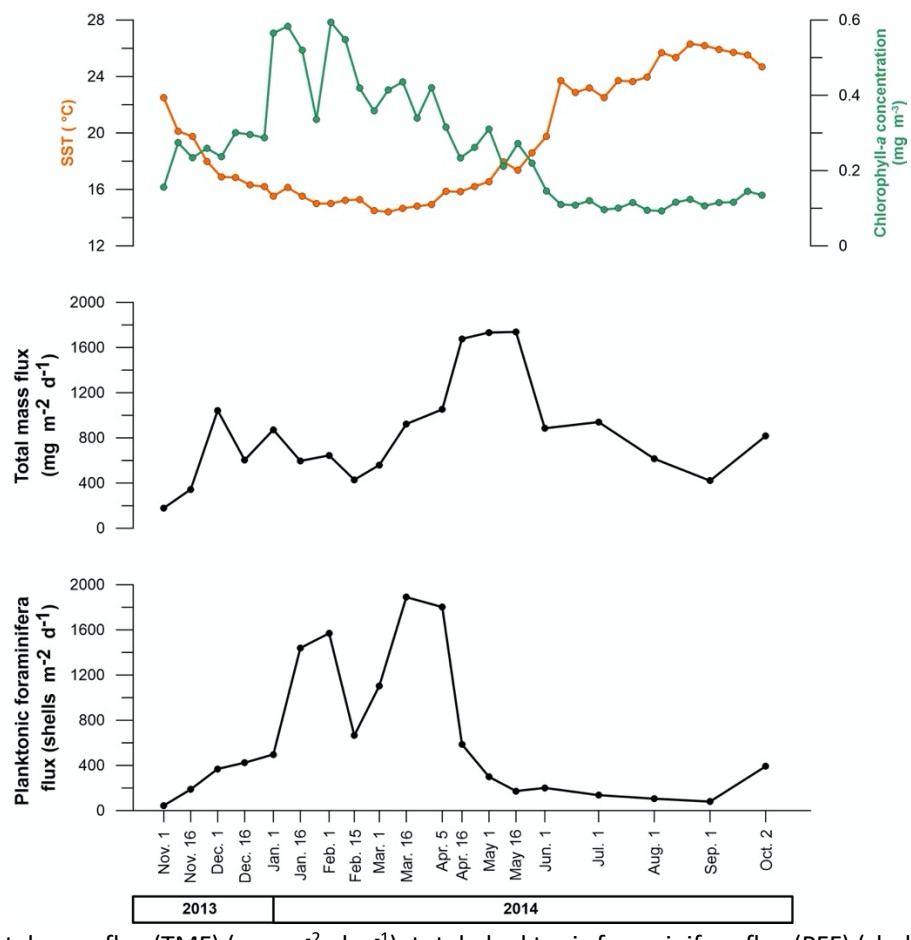

**Figure 2.** Total mass flux (TMF) (mg m$^{-2}$ day$^{-1}$), total planktonic foraminifera flux (PFF) (shells m$^{-2}$ day$^{-1}$),
SST (°C) and chlorophyll-*a* concentration (mg m$^{-3}$) changes between November 2013 and October 2014.

The mean total mass flux for the whole period of the study was 772.5 mg m$^{-2}$ d$^{-1}$, with a maximum
value of 1737.7 mg m$^{-2}$ d$^{-1}$ and a minimum value of 179.5 mg m$^{-2}$ d$^{-1}$ reached in mid-May 2014 and
November 2013 respectively (Figure 2). Higher total mass flux values were reached during spring
2014, while lower values appeared during both autumn 2013 and 2014.
Planktonic foraminifera mean flux across the interval studied was 629.8 shells m$^{-2}$ d$^{-1}$ with a
maximum value of 1889.9 shells m$^{-2}$ d$^{-1}$ and a minimum of 44.6 shells m$^{-2}$ d$^{-1}$ reached in mid-March
2014 and in November 2013 respectively. Higher values occurred during two periods, early spring
and winter 2014, while the lower ones occurred from late spring to fall 2014. Overall, the seasonal
mean values were 1194.3 shells m$^{-2}$ d$^{-1}$ for the winter period, 612.3 shells m$^{-2}$ d$^{-1}$ for spring, 283.5
shells m$^{-2}$ d$^{-1}$ for autumn and finally 107.2 shells m$^{-2}$ d$^{-1}$ for summer.
SST mean value was 19.2 °C and values ranged between a maximum of 26.1 and a minimum of 14.5
°C. The mean chlorophyll-*a* value was 0.27 mg m$^{-3}$, the maximum value displayed was 0.56 mg m$^{-3}$
while the minimum one was 0.09 mg m$^{-3}$ (Figure 2).

**4.3 Foraminifera species fluxes**


Overall, most of the planktonic foraminifera species collected by the trap exhibited either a uni-
modal or bi-modal flux distribution with a few exceptions (Figure 3).
*Globorotalia inflata* exhibited the highest fluxes of all species, with a mean flux of 368 shells m$^{-2}$ d$^{-1}$
throughout the record, with peak values in mid-March 2014 (1361 shells m$^{-2}$ d$^{-1}$) and minimum in
November 2013 (3 shells m$^{-2}$ d$^{-1}$). *G. truncatulinoides* was the second most important contributor
(mean of 114 shells m$^{-2}$ d$^{-1}$), with a maximum in mid-February and a minimum in November 2013
(368 and 3 shells m$^{-2}$ d$^{-1}$, respectively). *G. bulloides* was the third most important contributor to the
total planktonic foraminifera fluxes with a mean flux of 57.2 shells m$^{-2}$ d$^{-1}$ and maximum values
registered in April 2014 and minima in November 2013 (482 and 0 shells m$^{-2}$d$^{-1}$, respectively).
The remaining species displayed mean fluxes lower than 50 shells m$^{-2}$d$^{-1}$. *G. calida*, *G. ruber*, *G. ruber*
(pink), *G. rubescens* and *O. universa* mean fluxes were comprised between 10 and 20. Among these
species, *G. ruber* and *G. ruber* (pink) stood out and showed maximum fluxes of 66 shells m$^{-2}$ d$^{-1}$ in
February 2014 and 127 shells m$^{-2}$ d$^{-1}$ in October 2014, respectively. The remaining species, *G.*
*siphonifera*, *G. scitula*, *N. incompta* and *T. quinqueloba* mean and maximum fluxes were lower than
10 and 35 shells m$^{-2}$ d$^{-1}$, respectively, thereby representing a low contribution to the total
foraminifera fluxes.
Finally, it is worth noting that benthic foraminifera were also collected by the trap, displaying a mean
flux of 25 shells m$^{-2}$ d$^{-1}$. The peak contribution of these taxa was recorded in April 2014 (182 shells
m$^{-2}$ d$^{-1}$), and a minimum value in January 2014 (3 shells m$^{-2}$ d$^{-1}$). In terms of annualized foraminifera
flux, their contribution was only a 1.1% of the total foraminifera identified of which 75% was
recorded during April 2014 (Figure 6).

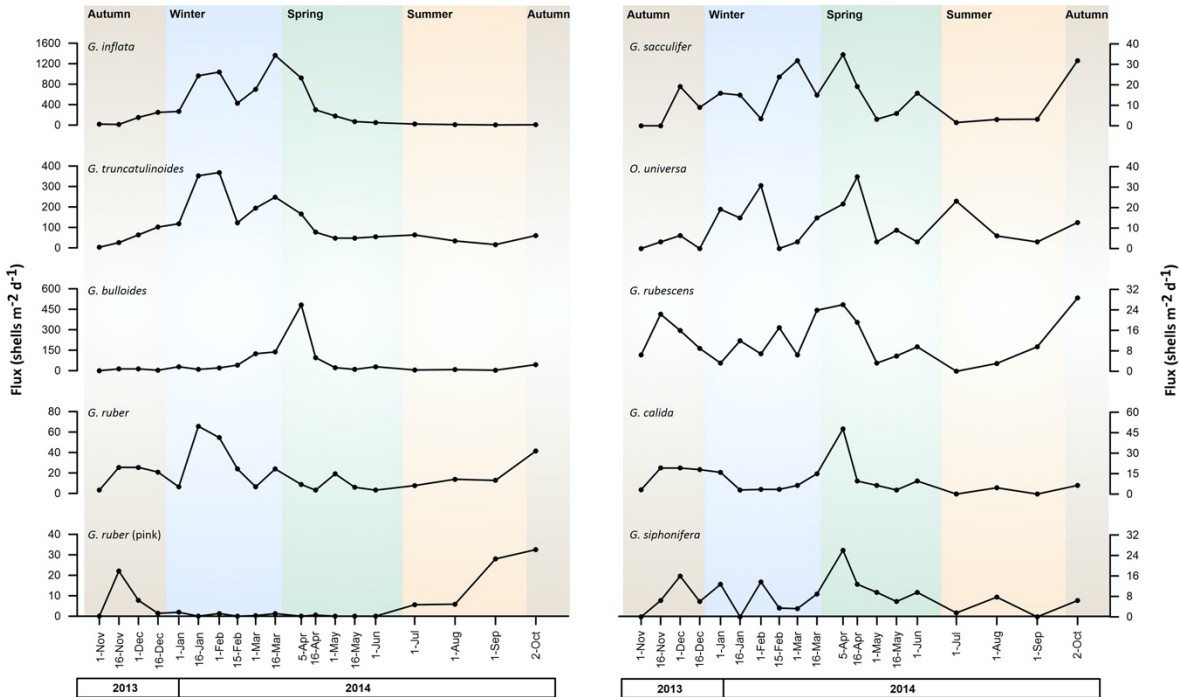

**Figure 3.** Planktonic foraminifera fluxes (shells m$^{-2}$ d$^{-1}$) from November 2013 to October 2014 of the 10 most abundant species identified. Note that the scale of the fluxes depend on the species. Background colour filling represents the different seasons: brown for autumn, blue for winter, green for spring and orange for summer.

The variations in relative abundance differed according to the species. Most of the species displayed a unimodal distribution across the studied interval (Supplementary Figure 3), with some exceptions such as *G. siphonifera*, *G. calida* or *G. ruber*. Overall, *G. inflata* dominated the association from late autumn until mid-spring. *G. truncatulinoides* relative abundance pattern was similar to that of *G. inflata*, with maximum values in autumn and late summer. In turn *G. bulloides*, displayed a pronounced seasonal change in its relative abundance reaching values up to 27% in early spring (April 2014) and dropping to about 5-8% in November 2014.

Overall, *G. inflata* is the only species that displayed its maximum mean relative abundance during winter: 64%. *G. siphonifera*, *G. sacculifer*, and *G. bulloides*, maximum mean relative abundances were reached during spring: 3%, 3.5%, 14% respectively. *G. calida*, *G. tenella*, *G. rubescens* and *N. incompta* maximum mean abundances appeared to be in autumn: 5.7%, 2.2%, 8% and 4.8% respectively. Finally, *G. ruber*, *G. ruber* (pink), *O. universa*, *G. truncatulinoides* and *G. scitula* maximum mean relative abundances were displayed in summer: 11.6%, 13.2%, 8.9%, 32.8% and 6.4% respectively (Supplementary Figure 3).

**4.4 Chlorophyll-*a* and SST impact on foraminifera fluxes**

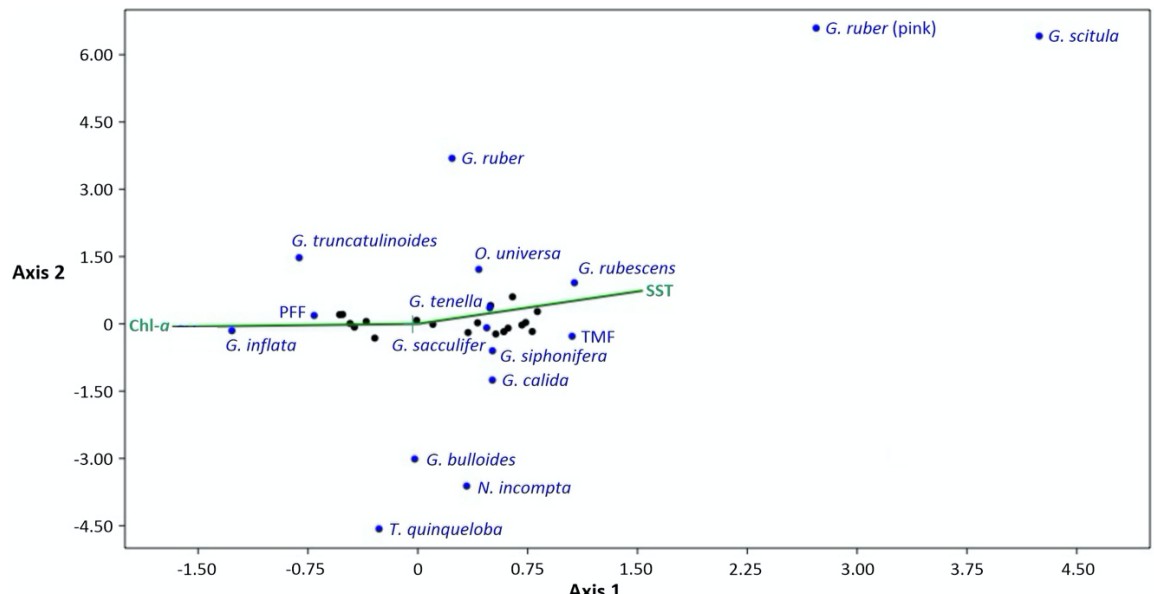

**Figure 4.** CCA analysis of all the planktonic foraminifera species flux with the SST (°C) and the chlorophyll-
*a* ("chl-*a*" in the CCA, in mg m$^{-3}$) as the explanatory variables. The total mass flux ("TMF") and planktonic
foraminifera flux ("PFF") are also included. Black dots represent the 19 sediment trap samples studied.
A CCA (see section 3.4) was carried out to characterize the impact of both the SST and the
chlorophyll-*a* on the planktonic foraminifera fluxes (Figure 4).
Axis 1 shows, overall, the differences between deep and surface dwellers. Total planktonic
foraminifera flux (PFF) and the fluxes of *G. inflata* and *G. truncatulinoides* are positively affected by
the chlorophyll-*a* concentration and negatively affected by the SST. On the other hand, *G. ruber*, *G.*
*ruber* (pink) and *G. scitula* fluxes showed an opposite pattern, being positively related with the SST
and negatively with the chlorophyll-*a* concentration. *O. universa*, *G. rubescens*, *G. tenella*, *G.*
*sacculifer*, *G. siphonifera* and *G. calida* fluxes are positively correlated with the SST and negatively
with chlorophyll-*a* concentration, nonetheless, the impact of these parameters is weaker compared
with the previous species. Finally, *G. bulloides*, *N. incompta* and *T. quinqueloba* fluxes are slightly
positively influenced by the chlorophyll-*a* concentration, however. Axis 2 tends to separate the
species between the different trophic regimes. Overall, it confirms that, in the one hand, *G. ruber*,
*G. ruber* (pink) and *G. scitula* display a strong negative correlation with chlorophyll-*a* and therefore
an affinity for oligotrophic and warm conditions; and on the other hand, shows that *G. bulloides*, *N.*
*incompta* and *T. quinqueloba* display a positive correlation with chlorophyll-*a* and eutrophic
conditions. Furthermore, *G. bulloides* flux shows a strong correlation with the latter two species:
0.89 and 0.83 ($p<0.05$).
**5. Discussion**
**5.1 Seasonal variations in the magnitude of planktonic foraminifera fluxes in the Sicily Channel**

The strong seasonality in the planktonic foraminifera fluxes registered by the trap is generally similar
in amplitude to previous studies in the Mediterranean (Bárcena et al., 2004; Rigual-Hernández et
al., 2012) and other temperate settings (Kuroyanagi and Kawahata, 2004; Wilke et al., 2009),
thereby suggesting the C01 record mainly reflects the temporal variations in planktonic foraminifera
abundance in the upper water column. Therefore, next, we discuss the influence of oceanographic
controls on the planktonic foraminifera fluxes.
Our data shows that, despite differences in the magnitude of their fluxes, most of the species
identified display their maximum flux during winter, winter/spring transition or spring (Figure 3)
thereby coinciding with the period of maximum algal biomass accumulation and coldest SSTs (Figure
2). The enhanced primary productivity during winter and spring is mostly related to an
intensification of the chlorophyll-*a* and nutrient richer MAW flow into the Eastern Mediterranean
basin (D'Ortenzio, 2009; Pinardi et al., 2015; Siokou-Frangou et al., 2010). Our CCA results (Figure 4)
show that, although the flux patterns increase during winter and spring, only the planktonic
foraminifera flux, *G. inflata*, *G. truncatulinoides* and arguably *G. bulloides* (further discussed below)
fluxes are negatively related to SSTs and positively with the chlorophyll-*a* concentration. The
dominance of the planktonic foraminifera fluxes by these three species and their affinity for
mesotrophic waters is not surprising as *G. inflata* and *G. truncatulinoides* are typically associated
with the MAW, winter water mixing events and hydrologic fronts in the western Mediterranean,
while *G. bulloides* is generally associated with eutrophic environments linked to upwelling
conditions (Azibeiro et al., 2023). Overall, these three taxa have been described to be dominant
during winter in various western regions of the Mediterranean, such as the Alboran Sea (Bárcena et
al., 2004; Hernández-Almeida et al., 2011), the Provençal basin and in the Gulf of Lions (Pujol and
Grazzini, 1995; Rigual-Hernández et al., 2012). Interestingly *G. inflata*, *G. truncatulinoides* and *G.*
*bulloides* are almost absent in the eastern part of the basin, most likely due to the low algal biomass
accumulation (Avnaim-Katav et al., 2020; Thunell, 1978).
Conversely, species such as *G. ruber*, *G. ruber* (pink), *G. scitula*, *G. rubescens* and *G. sacculifer* display
their maximum fluxes in summer or autumn (Figure 3). During the warm periods, summer and
autumn, the eastward advection of Atlantic waters in the Sicily Channel is weakened due to an
increased meandering of the ATC (Figure 1) and the local hydrography patterns (Béranger et al.,
2004), leading to a local water column stratification period which is also well documented in the
whole Mediterranean basin during summer (Siokou-Frangou et al., 2010). This translates into a
reduced MAW influence, and a larger influence of the LIW at intermediate depths (Astraldi et al.,
2002, 2001; Jouini et al., 2016). Therefore, the water column becomes warmer, saltier and more
nutrient depleted than the general conditions of the western basin (Gasparini et al., 2005; Navarro
et al., 2017; Siokou-Frangou et al., 2010) and provides the necessary environmental and
oceanographical configuration for eastern basins taxa to develop or being transported from the
easternmost part of the Mediterranean. Indeed, our CCA results (Figure 4) support these
observations (Figure 3). The latter species have been described to reach their maximum abundances
in the eastern part of the Mediterranean, specifically in the Ionian and Levantine basins during both
summer and autumn (Avnaim-Katav et al., 2020; Pujol and Grazzini, 1995).
Some species, such as *O. universa* or *G. calida*, do not display a clear flux pattern over the period
studied. CCA results suggest that these species have an affinity for warm and less productive
conditions. These taxa are considered widespread in the Mediterranean basin, although their
relative contributions are generally higher in the eastern part of the basin (Avnaim-Katav et al.,
2020; Pujol and Grazzini, 1995; Thunell, 1978). Lastly, it is important to note that the low number of
specimens of *G. falconensis*, *N. incompta*, *T. quinqueloba* and *G. tenella* found in our samples, makes
the estimation of shell fluxes for these species unreliable. These results are not surprising, since *N.*
*incompta* is mainly found in the northwestern part of the basin owing to cold and eutrophic
conditions (Azibeiro et al., 2023; Millot and Taupier-Letage, 2005) while *T. quinqueloba* has generally
been associated to cool Atlantic waters or cool marginal seas (Azibeiro et al., 2023).
In summary, planktonic foraminifera flux was maximum during winter and spring, coinciding with
the maximum seasonal eastward advection that brings MAW further east into the Sicily Channel.
These waters are less saline and nutrient enriched compared to the easternmost waters from the
Levantine basin*. G. inflata*, *G. truncatulinoides* and *G. bulloides* (the three most abundant species
that dominate the PFF), which are species described to come from the western basins, are probably
brought by the MAW and then dominate the planktonic foraminifera population. On the other hand,
during summer and autumn, the eastward advection weakens, allowing the LIW and AIS to
dominate the surface circulation due to the water column stratification and set favourable
conditions for eastern basin dominant taxa such as both morphotypes of *G. ruber*, *G. rubescens*, *G.*
*sacculifer*. This results in a significantly decreased planktonic foraminifera flux due to the absence
of western basin dominant species.

**5.2 Species succession, ecology and impact of the SST and chlorophyll-*a***

The time series of settling planktonic foraminifera reflects a diverse assemblage with species with
contrastingly different ecological preferences, encompassing a wide range of depth habitats and
diverse feeding strategies. Overall, the annual assemblage composition agrees well with previous
ship-board observations (Pujol and Grazzini, 1995) in the Channel of Sicily during VICOMED 1988
cruise, where *G. inflata*, *G. truncatulinoides* and *G. bulloides* were documented as the most
abundant taxa.
Next, we discuss the ecology of the most abundant species and the impact of chlorophyll-*a* and SST
on their distribution. We also discuss the foraminifera groups suggested by Jonkers and Kučera,
(2015), to explore their correlation with the previous parameters on an interannual scale. The latter
work proposed 3 groups: group 1 is formed by tropical and subtropical species, group 2 consists of
temperate to subpolar taxa, and group 3 represents the deep-dwelling species. These groups were
described as a result of the seasonal maximum fluxes timing of each species and their relationship
with both temperatures and nutrients (amongst other parameters) in different time-series across
the world ocean. Therefore, here we also used this grouping to compare and complete this
classification from a new time-series dataset.
*Globorotalia inflata* is the most abundant taxon in our samples. Our data shows that maximum
fluxes and relative abundances of this species are reached during winter and the winter-spring
transition (Figure 3). The relative abundances showed strong positive and negative significant (*p*
<0.05) correlations with the chlorophyll-*a* concentration and the SST: 0.808 and -0.896 respectively
(Figure 5). It is a non-spinose species and is considered a deep dweller (Hemleben et al., 1989;
Schiebel and Hemleben, 2017). Generally regarded as showing limited opportunistic behaviour and
it has been often associated with eddies and hydrological fronts (Chapman, 2010; Retailleau et al.,
2011). Concerning the Mediterranean, its maximum stocks and abundances have been recorded
along the southern margin of the western Mediterranean basin (Azibeiro et al., 2023), especially
during winter (Bárcena et al., 2004; Pujol and Grazzini, 1995; Rigual-Hernández et al., 2012); while
it is poorly represented in the eastern part, even absent in the Levantine basin (Avnaim-Katav et al.,
2020). As a consequence, *G. inflata* can be considered as a mesotrophic species, which is dominant
in regions with some degree of stratification of the water column and an intermediate amount of
nutrients and it has been used as a tracer of the Atlantic inflow in the Mediterranean basin (Azibeiro
et al., 2023), which agrees with the local hydrography in the Sicily Channel during winter and spring.
As *G. inflata* appeared in periods of cool and nutrient enriched waters (Figure 3), which coincide
with the periods of higher MAW influence in the Sicily Channel (Béranger et al., 2004), we consider
that our results further confirm *G. inflata* as tracer of the MAW in the Sicily Channel.
*Globorotalia truncatulinoides* is the second most abundant species in our record. However, our CCA
results suggest that the seasonal variations in *G. truncatulinoides* are not directly correlated with
either chlorophyll-*a* concentration or SSTs (r= -0.162 and 0.256, respectively, *p* >0.05) (Figure 5).
This highlights the fact that environmental controls other than the ones considered here may be
affecting its distribution. This taxon is a cosmopolitan species found in all major oceans (Schiebel
and Hemleben, 2017) and is considered a deep dweller with an affinity for water-mixing conditions
(Margaritelli et al., 2020; Schiebel and Hemleben, 2005). It is a non-spinose species with a complex
life cycle. In the Mediterranean, peak abundances of this species are found in the northwestern part
of the basin, where it represents a major component of the assemblages (Pujol and Grazzini, 1995;
Rigual-Hernández et al., 2012), while it is absent in the easternmost part of the basin (Avnaim-Katav
et al., 2020). This species has been documented to have a complex life cycle and reproductive
strategy. *G. truncatulinoides* has been described to reproduce once a year in the upper layers of the
water column, generally when the water mixing allows the migration of juvenile individuals to the
surface (Lohmann and Schweitzer, 1990; Schiebel et al., 2002). Then, adult individuals migrate
downward the water column and spend the rest of their life cycle (Rebotim et al., 2017; Schiebel
and Hemleben, 2005). Hence, we speculate that these complex migratory patterns may be playing
a role here. As its reproduction cycle is mainly controlled by the gametogenesis process, and as
described previously, it reproduces once a year (a slower rate than the majority of the planktonic
foraminifera species) (Schiebel and Hemleben, 2017), then, although different stages of its life cycle
could be affected by SST and chlorophyll-*a*, this is not necessarily registered by the sediment traps
in every stage of its growth.
*Globigerina bulloides* was the third most abundant planktonic foraminifera species identified here.
It is a surface to subsurface dweller and one of the most common species across the world ocean
(Schiebel and Hemleben, 2017). Interestingly, our analysis showed no significant correlation
between changes in *G. bulloides* relative abundance and chlorophyll-*a* concentration or SST (r= -
0.145 and -0.111 respectively, *p* >0.05). However, across the time span studied, this taxon showed
its maximum abundance and fluxes during relatively high chlorophyll-*a* and cool SST conditions
(Figure 3). This highlights that other environmental parameters than the ones considered here might
be playing a role in its distribution. It is a spinose species known for its opportunistic feeding strategy
(Schiebel et al., 2001) and affinity for upwelling and eutrophic environments (Azibeiro et al., 2023;
Bé et al., 1977). Within the Mediterranean Sea, it displays peak export fluxes to the deep sea in
areas of high productivity such as the Gulf of Lions and the Alboran Sea during the high productivity
period in late winter to spring (Azibeiro et al., 2023; Bárcena et al., 2004; Hernández-Almeida et al.,
2011; Rigual-Hernández et al., 2012), while few individuals are found in the eastern part of the
Mediterranean (Avnaim-Katav et al., 2020). We surmise that owing to its multiple trophic strategies
and its multi-diet characteristics, it could adapt and feed on varying chlorophyll-*a* concentrations.
Also, the lack of correlation with both parameters could be explained by the fact that this taxon is
associated with eutrophic conditions. In the Sicily Channel, the high productivity period ranges from
winter to spring, and the conditions allow deep mesotrophic dwellers (i.e. *G. inflata*) to dominate
the assemblage; while in summer and autumn, the upwelling setting brings oligotrophic conditions
that are not favourable for this species.
Generally, both *G. bulloides* and *G. truncatulinoides* fluxes and abundances are positively linked to
favourable food conditions and high-productivity environments. The first species tends to exhibit a
"bloom" strategy on short time scales, while the second species tends to be related to nutrient
advection zones in the Mediterranean Sea (Margaritelli et al., 2022). Furthermore, in the
Northwestern Mediterranean a previous study showed that the fluxes of these two species are
almost in phase (Rigual-Hernández et al., 2012). Interestingly, in the Sicily Channel, this relation is
not straightforward. In the Gulf of Lions, *G. bulloides* is the main species and shows the classical
"bloom" behaviour, while *G. truncatulinoides* pattern is more constant and its variations are more
gradual (Rigual-Hernández et al., 2012). Although the timing of the two species is different in our
record, the response of *G. truncatulinoides* is similar across the record. Furthermore, from a
productivity standpoint, the Sicily Channel is less productive than the Gulf of Lions (Siokou-Frangou
et al., 2010), which, in turn, does not benefit *G. bulloides* abundances and, as the upwelling in our
study zone is less pronounced than in other parts of the Mediterranean, the timing between the
two species is different. Additionally, the intensity of the upwelling in the central Mediterranean is
controlled by variations in the intensity of the LIW flowing to the western part of the basin (Astraldi
et al., 2001; Lermusiaux and Robinson, 2001; Pinardi et al., 2015), with higher intensity leading to
reduced upwelling and therefore, productivity. This could explain the lack of high abundance of *G.*
*bulloides* in our study region as the upwelling in the Sicily Channel is reduced compared to other
places in the Mediterranean (D'Ortenzio, 2009; Siokou-Frangou et al., 2010) and therefore, the
increase in productivity is diminished compared to other regions in which the productivity and the
abundance of *G. bulloides* are higher, such as the Alboran Sea (Bárcena et al., 2004). Therefore, we
consider that a combination of ecological preferences and oceanographic processes could explain
the lack of synchronicity between these two species fluxes and abundances.
*Globigerinoides ruber* and *G. ruber* (pink) were the fourth and fifth most abundant species in our
samples (Table 1). Our correlation analyses showed a significant positive effect of SST (r= 0.803 and
0.678, *p* <0.05) and a significant negative effect of chlorophyll-*a* (r= -0.567 and -0.464 respectively,
*p* <0.05) on both *G. ruber* and *G. ruber* (pink) respectively (Figure 5). These species have been
described as tropical to subtropical taxa, with an affinity for oligotrophic and stratified waters (Bé
et al., 1977). Both of these species are among the shallowest dwellers of the extant planktonic
foraminifera species and are considered one of the most adaptable to varying surface water
conditions (Kemle-von Mücke and Oberhänsli, 1999; Schiebel and Hemleben, 2017). Due to its
temperature and salinity limits for food acceptance, the white variety is one of the most studied
foraminifera species in culture experiments, which highlight their euryhaline and eurythermal life
cycle (Bijma et al., 1990; Lombard et al., 2009). In today's ocean, the white variety is substantially
more abundant than the pink one (Schiebel and Hemleben, 2017). In the case of the Mediterranean
basin, *G. ruber* is generally associated with warm and oligotrophic waters (Pujol and Grazzini, 1995)
and is abundant in the eastern oligotrophic basin, where it dominates the assemblages in the
Levantine basin during spring and fall (Avnaim-Katav et al., 2020). However, although present in the
western basin, its abundance is much lower in the Gulf of Lions (Rigual-Hernández et al., 2012) and
in the Alboran Sea (Bárcena et al., 2004). Overall, the correlation data agrees with the previous work
that linked *G. ruber* (both varieties) to warm and oligotrophic conditions generally displayed during
a higher stratification of the water column (Schiebel et al., 2004). As this species is mostly abundant
in the eastern part of the Mediterranean , it should be expected that the LIW, when it dominates
the circulation during summer and autumn, brings this species along with other oligotrophic taxa.
However, fluxes (Figure 3) and relative abundance data (supplementary Figure 3) showed that this
species maximum appearances were recorded during winter, coincidently with *G. inflata* and *G.
truncatulinoides*. Therefore, the winter recorded in our dataset showed favorable conditions for
both deep mesotrophic dwellers and oligotrophic species such as *G. ruber*. We interpret this pattern
as a reduced influence of the MAW during winter in the Sicily Channel that could lead to slightly
warmer than usual surface conditions that favor the stratification and hence, the *G. ruber*
abundances.  Concerning *G. ruber* (pink), as its fluxes and abundances were higher during summer,
and it is mainly identified in the eastern part of the Mediterranean as well, we consider that the LIW
influence bring this species in the Sicily Channel.
According to Jonkers and Kučera, (2015), the foraminifera fluxes can be predicted on a seasonal
scale for three different groups of planktonic foraminifera. Following this approach, we explore the
relative abundance of these three aggrupations to document if these correlate with both SST and
chlorophyll-*a* concentration (see Supplementary Table 1) on the period covered by the sediment
trap (Figure 5). The first group (group 1) consists of both *G. ruber* varieties, *G. sacculifer*, *O. universa*,
*G. siphonifera*, *G. rubescens* and *G. tenella*. The second group (group 2) is formed by *G. bulloides*, *T.
quinqueloba*, *N. incompta*, *G. scitula* and *G. calida*. In our record, however, either *G. bulloides* or *G.
calida* displayed a similar trend, and the remaining three species abundance was <1.5%, making any
significant assumption difficult (Table 1). The third (group 3) is composed by the deep dwellers *G.
inflata* and *G. truncatulinoides*. Group 1 showed a strong and significant positive correlation with
the SST (Figure 5) and a negative with the chlorophyll-*a* (r= 0.828 and -0.668 respectively, *p* <0.05,
see Supplementary Table 1). This is not surprising as the majority of the group is formed by species
not only considered tropical but also well adapted to oligotrophic and nutrient impoverished
environments (Chapman, 2010; Hemleben et al., 1989; Schiebel and Hemleben, 2017). In addition,
most components of this group are symbiont bearing species (Takagi et al., 2019), which have been
described to be more adapted to nutrient depleted and oligotrophic conditions. Group 2 on the
other hand did not show any strong correlation to either SST and chlorophyll-*a* concentration,
although a significant negative correlation was displayed between the group abundances and the
latter parameter (r= -0.525, see Supplementary Table 1). This result is not surprising as the main
component of this group is *G. bulloides*, which previously showed a lack of correlation with both SST
and chlorophyll-*a*, while the remaining species of this group were taxa that tend to be outnumbered
by more opportunistic species (i.e. *N. incompta* and *T. quinqueloba*) (Kuroyanagi and Kawahata,
2004; Schiebel, 2002). Also, the overall abundance of these taxa was very low in our samples
compared to the other two groups, which in turn could affect the correlation results. Here we
propose that the mesotrophic conditions of the Sicily Channel developed during the relatively high
productivity period are not favourable enough for the development of the taxa comprised in group
2. Finally, group 3 displayed a strong and significant positive correlation with chlorophyll-*a*
concentration (r= 0.771, *p* <0.05), which is an expected trend according to the affinity showed to
mesotrophic conditions by the two species that constitute this group, however, as compared to
Jonkers and Kučera, (2015), we showed a strong and significant negative correlation of these two
species abundances with the SST (Figure 5). The latter work stated that the cycles of these species
were independent of the temperature changes, however, these two species tend to be used as
tracers of cool and deep mesotrophic waters in the Mediterranean, generally associated with the
MAW (Azibeiro et al., 2023).
In summary, our data showed that in the Sicily Channel, the three major ecological groups proposed
by Jonkers and Kučera, (2015), exhibited a different response to environmental variability. Overall,
groups 1 and 3 showed significant correlation with the latter parameters and were in accordance
with their corresponding species ecologies. However, group 2 did not show any significant
correlation, which we interpreted as the result of very low abundances of the taxa comprised within
this group. This translates into the dominance of group 1 during summer and autumn when
oligotrophic and warm eastern waters dominate the water column, while the mesotrophic taxa from
group 3 dominate during winter and spring, coincidently with higher primary productivity, yet not
eutrophic enough for the opportunistic taxa comprised in the group 2, which is less well represented
in the Sicily Channel.

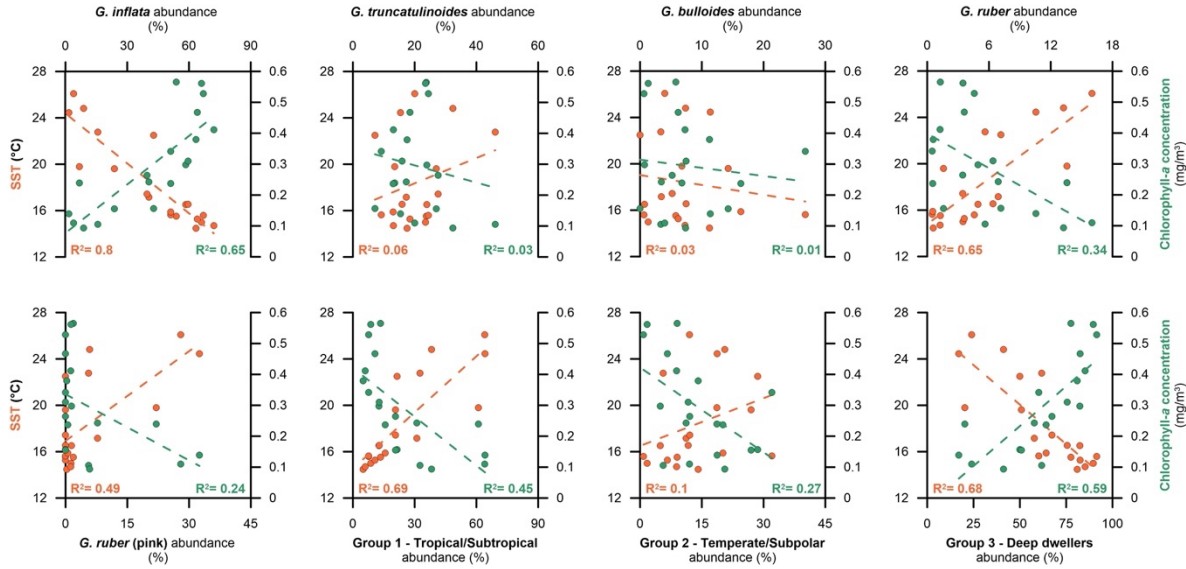


 **Figure 5.** SST and chlorophyll-*a* concentration against the relative abundance of the five most abundant
species and the three ecological groups proposed by Jonkers and Kučera (2015). Orange dots stand for
SST while the green ones correspond to chlorophyll-*a*.
**5.3. Influence of the hydrodynamic conditions on the planktonic foraminifera assemblage**

A possible source of variability between the living foraminifera assemblages and those collected by
the trap could be the preferential transport of certain species by the currents as well as differences
in the sinking rates between species. Typically, deep dwelling species produce heavier shells that
the surface dwelling ones (Zarkogiannis et al., 2022). Theoretically, lighter species are easier to
remobilize than the heavier ones, however, if the current is strong enough, lighter species could
travel far away while heavier species could be reworked in the vicinity of their deposition zone. *G.*
*truncatulinoides* is among the heaviest planktonic foraminifera species (Beer et al., 2010; Béjard et
al., 2023). Therefore, if the current is strong enough, it could be resuspended and be recorded by
the sediment trap. The record in the seabed sediment (see section 5.5) showed that *G.*
*truncatulinoides* was more abundant in the settling particles from the C01 mooring line (Figure 8),
and according to the winnowing theory, *G. inflata* should follow a similar pattern as it also a heavy
species (Zarkogiannis et al., 2022). However, surface data (Mallo et al., 2017) showed that the latter
is also the dominant species in the BONGO nets (see section 5.5). Furthermore, Takahashi and Be,
(1984) presented the data about the sinking speeds of different planktonic foraminifera species. As
an example, *G. inflata* showed a sinking speed of 500 m per day, compared to 330 m per day for *G.*
*bulloides*. These different sinking rates applied in a water column of around 450 m suggest that the
likely origins of the planktonic foraminifera collected by the traps must be similar and are insufficient
to generate discrepancies between the foraminifera assemblages living in the upper water column
and those collected by the trap.

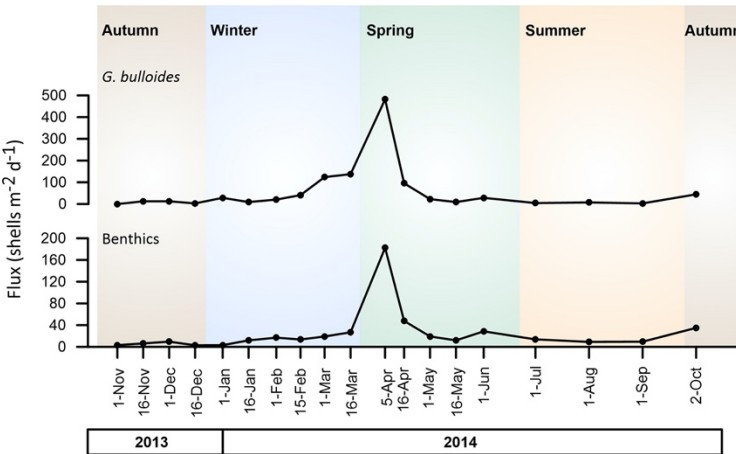

**Figure 6.** *G. bulloides* and benthic foraminifera fluxes (shells m$^{-2}$ d$^{-1}$) between November 2013 and
October 2014.

The identification of benthic foraminifera individuals highlights suggest an impact of the
hydrodynamic conditions on the settling particles populations. The main species identified were *T.*
*saggitula* spp. and *B. marginata* (Plate 1) along with a small number of *Uvigerina mediterranea* and
*Lagenina striata*. These taxa are considered infaunal species, i.e. they live buried in the sediment
(Balestra et al., 2017; Milker and Schmiedl, 2012) and are commonly found in continental shelves
and slopes. Overall, benthic foraminifera accounted only for a mean of 3.4% of the total foraminifera
identified in the C01 settling particles (Table 1) and the percentage of planktonic oscillated between
89 and 99.4%. Most of the annual benthic fluxes occurred during April, when a total of 80% of the
annual benthic foraminifera fluxes were recorded (Figure 6). As described previously, the Sicily
Channel hydrography is complex from both a vertical and seasonal point of view (Astraldi et al.,
2001; Garcia-Solsona et al., 2020; Incarbona et al., 2011; Pinardi et al., 2015; Schroeder et al., 2017).
In the Sicily Channel, the tidal and subtidal current speed is known to reach maximum annual values
during the spring period (Gasparini et al., 2004) which could be invoked as a possible source of
sediment resuspension including benthic species. This has also been observed in different parts of
the Mediterranean (Grifoll et al., 2019). Indeed, in our record, the highest benthic foraminifera
fluxes were collected during spring (Figure 6), i.e. the period of peak current intensity in the Channel.
Coincidently, it also showed the highest fluxes of *G. bulloides* (Figure 3), which is the third most
abundant species in our record (Table 1). Interestingly, this species annual flux distribution showed
no correlation with either the SST nor the chlorophyll-*a* (Figure 5). These observations, coupled with
the fact that the fluxes of *G. bulloides* and the benthic foraminifera were positively and significantly
correlated (r= 0.89, *p*<0.05), suggest that benthic species were resuspended, being caught at 40 m
of water depth by our sediment trap. Furthermore, a low number of detritic debris, such as mica
flakes, were identified in the samples that contained the highest number of benthic foraminifera
(April 2014), which again suggest a secondary influence of resuspended sediments in the sediment
trap record in specific intervals of the annual cycle. However, no such relationship has been
identified with the other species that did not show any correlation with the previous environmental
parameters: *G. truncatulinoides*. Consequently, we hereby propose that *G. bulloides* distribution
and abundances are blurred in specific intervals by the resuspension of sea floor sediments. Finally,
the increase of *G. bulloides* abundance and fluxes that has been identified coincidently with a higher
number of benthic foraminifera during early April could lead to the interpretation that the benthic
foraminifera are the result of the intensification of the MAW. However, as the presence of the
benthic foraminifera is patchy and not constant, we do not consider their presence is ruled out as a
reliable proxy for the MAW/LIW intensity. Therefore, it can be concluded that the C01 sediment
trap mainly records a pelagic signal  with a secondary influence of resuspended sediments.

**5.4 Geographical variability in the magnitude and composition of planktonic foraminifera fluxes**
**across the Mediterranean**

The comparison of the settling planktonic foraminifera assemblage from the Sicily Channel with the
ones retrieved from different parts of the Mediterranean offers a unique opportunity to provide
further insight into the central Mediterranean dynamics and ecology of this group.
As stated previously, the planktonic foraminifera flux in the Sicily Channel was higher from mid-
January to mid-March, which coincided with the highest chlorophyll concentrations and the coolest
SST recorded (Figure 2). This seasonality is similar to the one observed in the Gulf of Lions, where
the planktonic foraminifera flux reached its highest values from mid-February to mid-March during
different years (Rigual-Hernández et al., 2012). Although slightly different, the planktonic
foraminifera fluxes patterns from both the Levantine basin and the Alboran Sea also displayed
maximum values between mid-February to mid-March and mid-January to mid-February
respectively (Avnaim-Katav et al., 2020; Hernández-Almeida et al., 2011). However, the magnitude
of the planktonic foraminifera flux values displayed some differences between the sites (see
Supplementary Figure 2). Overall, for the Sicily Channel, values ranged between 0-1889 shells $m^{-2}$ $d^{-1}$
with a mean value of 629 shells $m^{-2}$ $d^{-1}$. These values were comparable to the ones from the Gulf
of Lions: 0-2114 and 4268 shells $m^{-2}$ $d^{-1}$ with a mean value of 225.4 in Planier sediment trap to 419
shells $m^{-2}$ $d^{-1}$ in Lacaze-Duthiers sediment trap (Figure 7). On the other hand, the Levantine basin
values were lower: 0-429 shells $m^{-2}$ $d^{-1}$, with a mean value of 93 shells $m^{-2}$ $d^{-1}$. Finally, the highest
values belonged to the Alboran Sea: 0-6000 shells $m^{-2}$ $d^{-1}$ with a mean value of 783 to 970 shells $m^{-2}$ $d^{-1}$
depending on the gyres. Note that the planktonic foraminifera flux values from the Levantine
basin used here represent the foraminifera shells from the >125 $\mu$m fraction, which highlights the
fact that compared to the >150 $\mu$m, the flux values should be even lower. The corresponding
chlorophyll-*a* values registered in the latter sites were 0.2-0.65 mg $m^{-3}$ for the Sicily Channel (Figure
5), 0.25-0.85 mg $m^{-3}$ for the Gulf of Lions (0-0.65 mg $m^{-3}$ in the Planier site, 0.25-0.85 mg $m^{-3}$ for
Lacaze-Duthiers) (Rigual-Hernández et al., 2012), 0.02-0.4 mg $m^{-3}$ for the Levantine basin (Avnaim-
Katav et al., 2020) and 0.1-1.2 mg $m^{-3}$ in the Alboran Sea (Hernández-Almeida et al., 2011), indicating
a similar productivity in terms of chlorophyll-a between the Gulf of Lions and the Sicily Channel. In
addition, here we calculated an annualized planktonic foraminifera flux (section 3.4) for each of the
6 sites compared here (Figure 7). Overall, the highest annualized fluxes were displayed in the
Alboran Sea (Figure 7): around $3x10^5$ and $4.4x10^5$ shells $m^{-2}$ $y^{-1}$, while the lowest one was displayed
in the Levantine Basin: a little over 30000 shells $m^{-2}$ $y^{-1}$ (Figure 7). The Gulf of Lions and the Sicily
Channel displayed comparable annualized fluxes although higher for the latter: around $1.5x10^5$ and
$1.85x10^5$ shells $m^{-2}$ $y^{-1}$ respectively. Note that PLA site values were significantly lower: around $7x10^4$
shells $m^{-2}$ $y^{-1}$ (Figure 7). Previous work showed that these planktonic foraminifera patterns were
mainly linked to specific regional oceanographic processes. First of all, the Levantine basin is well
known for being an ultra-oligotrophic region and being the warmest and saltiest of the
Mediterranean basins (Ozer et al., 2017), mainly due to the W-E anti-estuarine circulation. On the
other hand, the Gulf of Lions is considered an exception to the general oligotrophy of the
Mediterranean. Seasonal vertical mixing phenomenon occurs in winter, generated by cold winds.
This winter mixing recharges the surface waters with nutrients, allowing a winter/spring productivity
bloom (Durrieu de Madron et al., 2013; Houpert et al., 2016). Finally, the Alboran Sea is a transitional
region between the Atlantic Ocean and the Mediterranean Sea (Hernández-Almeida et al., 2011),
and unlike the latter, is not an oligotrophic region due to the two systems of high productivity
related to the gyres generated by an intense westerlies activity, which allow nutrients enriched
(compared to the resident waters) Atlantic waters to spread into the Mediterranean. This results in
an enhanced primary productivity period from November to March. According to the PFF patterns
displayed in this study, the Sicily Channel presents similar values and fluxes distributions to the Gulf
of Lions, however, its oceanographic circulation is significantly different from the latter. These
observations agree with the work of Mallo et al., (2017) carried out with plankton tows in the whole
Mediterranean basin. The latter work found that the Alboran Sea displayed the highest standing
stocks of planktonic foraminifera, while the easternmost part of the Mediterranean showed the
minimum values. Also, the Gulf of Lions and the Channel of Sicily displayed similar stocks, although
slightly superior for the Channel of Sicily.

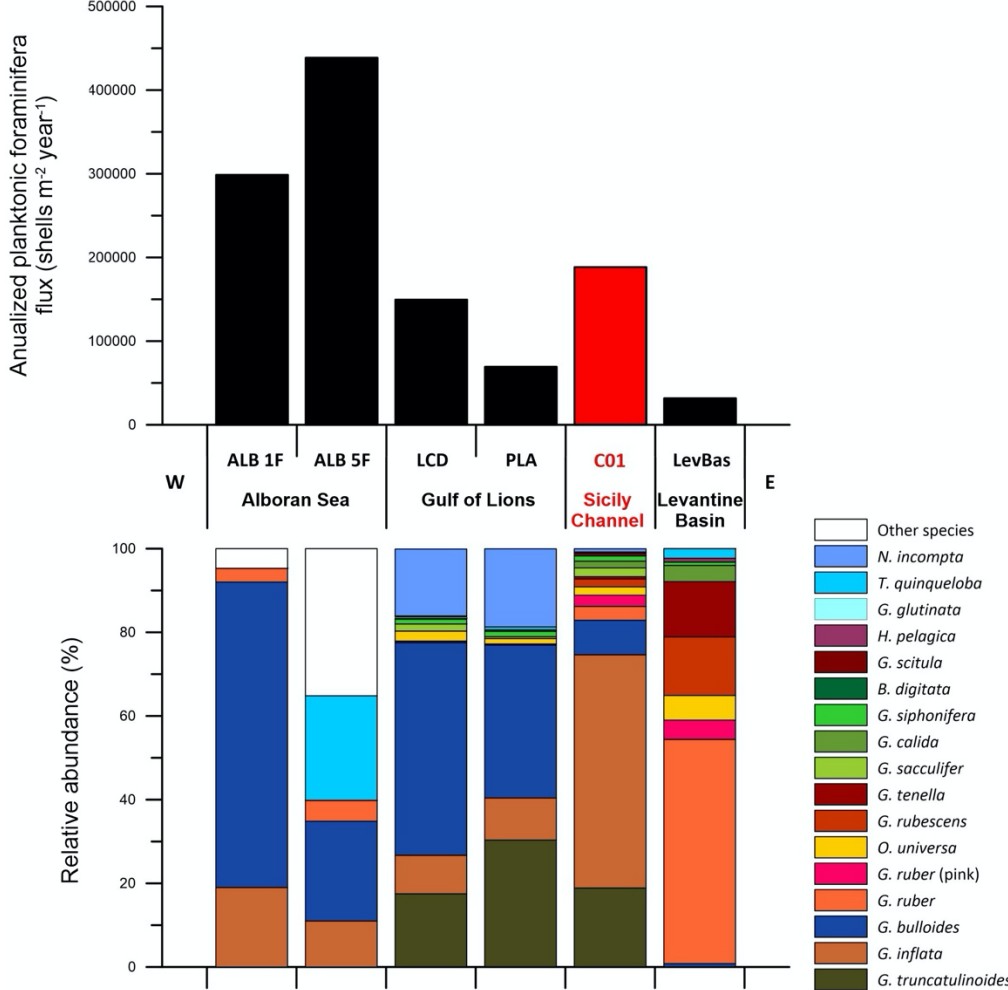

**Figure 7.** Comparison of the annualized (see section 3.4) planktonic foraminifera flux and the relative
abundance of each species identified in different time-series across the Mediterranean Sea (see section
3.5). The data from the Sicily Channel (C01) is depicted in red. Note that the Levantine Basin (LevBas)
dataset covers the >125 μm fraction. Other species (white bar) in the Alboran Sea corresponds to any
species different from the main 4 taxons identified in Bárcena et al., (2004) and Hernández-Almeida et
al., (2011).
Concerning the species composition, we identified 15 planktonic foraminifera species in the Sicily
Channel, which is a similar species number to the one from the Gulf of Lions (14 species) and higher
than in the Levantine basin (10 different species). The Sicily Channel site displayed the highest
planktonic foraminifera assemblage diversity among the three sites compared: a mean 1-D and S/W
index of 0.68 and 1.57 respectively. (Table 2). Interestingly, despite showing a similar number of
different species, the Gulf of Lions displayed the lowest diversity values, especially for the PLA site:
mean 1-D of 0.55 and mean H/W of 1.08, while the LCD site 1-D and h/w were 0.58 and 1.15

respectively. These observations highlight that, although the annualized planktonic foraminifera flux was similar between the Gulf of Lions (for the LCD site) and the Sicily Channel (Figure 7), the assemblage in the latter site was significantly more diverse regarding species composition. The composition of the annual planktonic foraminifera population of the different species showed some differences between the sites compared here. In the Levantine basin, the majority of the planktonic foraminifera population consisted of surface symbiont bearing species such as *G. ruber*, *G. ruber* (pink), *G. rubescens*, *G. tenella*, *O. universa*, which are well adapted to the ultra-oligotrophic conditions (Lombard et al., 2011; Schiebel and Hemleben, 2017). The latter species represented 96% of the total planktonic foraminifera in the Levantine basin, while the same species in the Sicily Channel accounted for around 10% of the total individuals (Figure 7). Note that both *G. rubescens* and *G. tenella* are considered small-sized species (Chernihovsky et al., 2023) and their adult size is often smaller than 150 µm, so it is possible that some individuals of those species may not be recorded in our data. On the other hand, in the Gulf of Lions, the four main species were *G. bulloides*, *N. incompta*, *G. inflata* and *G. truncatulinoides*, which represented 88 to 95% of the total planktonic foraminifera (Rigual-Hernández et al., 2012). These species tend to be associated with eutrophic to mesotrophic environments which coincides with the Gulf of Lions locally enhanced primary productivity conditions. In the Sicily Channel, the same species accounted for 83% of the total individuals, and, except for *N. incompta*, the remaining three species were also the most abundant in our samples.

**Table 2.** Inverse Simpson (1-H) and Shannon-Weiner indexes mean, standard deviation ("Stan. Dev.") and maximum values for the two Gulf of Lions sites (PLA and LCD), the Sicily Channel (C01, this study) and the Levantine Basin (LevBas).

| | Gulf of Lions | | Sicily Channel | Levantine Basin |
|---|---|---|---|---|
| | LCD | PLA | C01 | LevBas |
| **Simpson 1-H** | | | | |
| Mean | 0.581 | 0.553 | 0.681 | 0.615 |
| Stan. Dev. | 0.168 | 0.180 | 0.132 | 0.144 |
| Max | 0.802 | 0.781 | 0.872 | 0.804 |
| **Shannon H/W** | | | | |
| Mean | 1.151 | 1.078 | 1.572 | 1.230 |
| Stan. Dev. | 0.359 | 0.375 | 0.398 | 0.316 |
| Max | 1.789 | 1.630 | 2.188 | 1.759 |

Considering the planktonic foraminifera fluxes patterns, the species diversity and the planktonic foraminifera most abundant species from each of the three Mediterranean time-series with which we compared our data, we interpret that, from a planktonic foraminifera population point of view, the Sicily Channel could be considered as a transition zone and a biological corridor between the western and eastern basins.

Finally, to put our data into a global context, here we compare our dataset with planktonic
foraminifera data from the same size fraction retrieved in the Gulf of Mexico, high latitudes North
Atlantic and gyres region of the North Atlantic Ocean. In the northern Gulf of Mexico, from 2008 to
2010, the >150 μm PFF was comprised between 0 and slightly over 800 shells $m^{-2} d^{-1}$, with a mean
value of around 250 shells $m^{-2} d^{-1}$ (Poore et al., 2013). A total of 12 species were identified, with *G.*
*truncatulinoides*, *G. ruber* (pink) and *N. dutertrei* as the most abundant species recorded. On the
other hand, in the North and high-latitudes Atlantic Ocean, Wolfteich (1994), showed that the PFF
oscillated between 0 and around 5000 shells $m^{-2} d^{-1}$ for a mean value of 800 shells $m^{-2} d^{-1}$, while *G.*
*bulloides* and *N. incompta* were the most abundant species. Although the latter work only focused
on the most abundant species, additional work has documented more than 20 species in the vicinity
of the North-Atlantic gyres (Salmon et al., 2015), but around only three to four in the high latitudes.
This highlights that, from a planktonic foraminifera population point of view on a wider scale, the
Sicily Channel displayed a higher planktonic foraminifera flux and species richness compared to the
tropical to subtropical Gulf of Mexico and to the high latitudes of the North Atlantic, but lower values
compared to the North Atlantic gyres region.
**5.5 Recent planktonic foraminifera assemblage comparison with seabed sediment**
The Mediterranean Sea is often referred to as a climate change hotspot and a "laboratory basin" where
many global environmental trends are amplified (Bethoux et al., 1999). In particular, ocean warming is
expected to exceed the global average (Hassoun et al., 2022, 2015; Lazzari et al., 2014) while it is
considered a specially sensitive zone of the ocean to acidification due to the fast turnover of its waters
and penetration of anthropogenic $CO_2$ (Bethoux et al., 1999; Schneider et al., 2007). One of the main
questions about planktonic foraminifera concerns the way they are going to react to the ongoing climate
change in the global ocean (Jonkers and Kučera, 2015; Schiebel and Hemleben, 2017). Previous work
suggests that global communities of planktonic foraminifera have already been affected by
environmental change since the onset of industrialization (Jonkers et al., 2019). Moreover, recent work
has shown that the calcification of several planktonic foraminifera species has decreased during the
industrial era in the northwestern Mediterranean (Béjard et al., 2023). Therefore, here we aim to assess
if modern planktonic foraminifera communities dwelling in the Sicily Channel differ from their pre-
industrial counterparts. To do so, next, we compare the annual integrated assemblages collected by the
sediment trap in the C01 mooring line with the ones from a set of core-tops, two box-cores and BONGO
nets retrieved in the vicinity of the studied zone (see Section 3.5).
As planktonic foraminifera are a group of calcifying plankton, when comparing sediment trap and
seabed sediment data, the possible role of calcite dissolution must be discussed. Firstly, the
Mediterranean Sea is supersaturated with respect to calcite (Álvarez et al., 2014; Millero et al., 1979)
and the depth of the studied material is substantially shallower than the calcite saturation horizon
(Álvarez et al., 2014). Secondly, recent work suggests that calcite experiences little to negligible
changes in the water column and burial in recent sediments (Béjard et al., 2023; Pallacks et al.,
2023). All this evidence suggests that dissolution played a negligible role in the preservation of
planktonic foraminifera preserved in the sediment record in the study region.
The core-tops used for comparison were part of the MARGO database (see Section 3.5 for more
details). Note that the MARGO sites 3735 to 3739 seabed sediment was taken using a trigger-weight
corer (Thunell, 1978). However, samples 3658, 3672 and 3673 were retrieved using a piston corer
(Hayes et al., 2005). Generally, sampling with the trigger-weight method is considered to retrieve
less mixed and disturbed sediment than the piston or box corer sampling methods (Skinner and
McCave, 2003; Wu et al., 2020). Therefore, the foraminifera assemblages from the core-tops may
likely represent a mix of Holocene populations rather than exclusively modern assemblages.
Although the lack of dating control makes it impossible to determine the exact date of the core top
assemblages.
The sites 342 and 407, studied by Incarbona et al., (2019), were retrieved with a box-corer. A total
of 23 and 24 samples were analyzed in the latter work, respectively. The advantage of comparing
the C01 assemblages with those of Incarbona et al., (2019) is the availability of high resolution $^{210}$Pb
chronology. The ages ranged from 1718 to 1962 CE for site 342 and from 1558 to 1994 CE for site
407. Therefore, here we present a comparison with the mean relative abundance of the main
planktonic foraminifera species from all the samples available (Figure 8).
Finally, to provide a more complete snapshot of the surface assemblages, we also include the
abundances from Mallo et al., (2017) that were collected with a BONGO net during spring 2013 in
the axis of the Sicily Channel (Figure 8).
In terms of planktonic foraminifera assemblage composition, major differences were observed
between the different seabed sediments datasets (Figure 8). Overall, the settling population from
the C01 mooring line appeared to be closer to the assemblages from sites 342 and 407 (Figure 8)
than to the mean from the MARGO database (see Supplementary data). The most evident
observation relies on the shift of the dominant species when comparing the settling population with
the sites 342, 407, the BONGO net and the core-top assemblages (Figure 8). As described previously,
*G. inflata* dominated the assemblages collected by the sediment trap (Table 1). This is also the case
for the sites 342 and 407 and the BONGO net (Figure 8). However, *G. bulloides* was the best-
represented species in the core-tops from the MARGO database. Also, the second most abundant
species varied across the datasets: *G. ruber* in the sites 342 and 407, *O. universa* in the BONGO nets
and *G. inflata* in the MARGO core-tops, with abundances around 27-29, 29 and 27.5%, respectively.
Interestingly, *G. truncatulinoides* abundance was significantly lower in the seabed datasets and
absent in the BONGO nets, highlighting the deep aspect of its ecology (Figure 8). On the other hand,
the "other species" category, which consists of minor taxa such as *G. rubescens, G. siphonifera* and
*G. calida* (amongst others) played a more significant role in the MARGO core-tops and BONGO nets
assemblages, reaching abundances up to 26% (Figure 8), while in the sites 342 and 407, these
species abundances are similar to those of the sediment trap.
These results lead to several observations. Firstly, concerning the seabed sediment comparison, the
sediment trap assemblage is closer to the sites 342 and 407 than to the MARGO database core-tops.
The comparison with the surface BONGO nets shows that, although the dominant species are the
same (i.e. *G. inflata*), the influence of *O. universa* and the overall diversity is less important in surface
waters. This highlights the complexity of the Sicily Channel configuration and the differences
between the surface (BONGO nets), the water column (sediment trap) and the seabed sediment
(MARGO database and sites 342 and 407) regarding the planktonic foraminifera populations.
Secondly, the seabed sediment planktonic foraminifera populations showed a reduced influence of
deep-dwelling species (excepting for *G. inflata* in sites 342 and 407) and a more pronounced
influence of both eutrophic and oligotrophic species. These eutrophic species (such as *G. bulloides*
but also *N. incompta*) are associated with MAW and western basins in the modern Mediterranean
Sea, while the more oligotrophic taxa (*G. ruber*, *G. rubescens*, *G. calida*…) are considered to be
abundant in the easternmost part of the basin (Azibeiro et al., 2023). As noted previously, although
the settling assemblage differs to the ones from the seabed sediment, it is more similar to the sites
342 and 407 than to the MARGO database core-tops. Also, the [210]Pb chronology available for sites
342 and 407 covers the years 1558 to 1994 CE (Incarbona et al., 2019). A possible interpretation of
these results is that the MAW influence into the basin may have shifted. Instead of bringing rich and
eutrophic waters that would allow the development of opportunistic species, it nowadays brings
more mesotrophic water masses that favour the development of deep dwellers in the Sicily Channel.
On the other hand, this could also lead to the assumption of a reduced eastward and LIW influence
in the present day as seen by the significantly lower abundance of oligotrophic species in the settling
assemblages. Also, a change in the environmental conditions could lead to the increase of deep
dwellers in substitution of eutrophic species such as *G. bulloides*. As described previously, the
Mediterranean Sea has already been described as a climate change "hotspot", therefore the already
documented ocean warming and the consequent stratification (Malanotte-Rizzoli et al., 2014;
Siokou-Frangou et al., 2010) could have led to unfavorable conditions for several taxa. A decrease
in the primary production might have caused a shift in the dominance of the opportunistic *G.*
*bulloides* by *G. inflata*. As described previously, *G. bulloides* shows a high affinity for high
productivity environments, while deep dwellers such as *G. inflata* and *G. truncatulinoides* tend to
prefer mesotrophic and stratified waters. Finally, note that the high abundance of *G. bulloides* in
the seabed sediment could also be the result of a punctual high productivity events. In the Alboran
Sea, during upwelling events, big amounts of *G. bulloides* are deposited in the seabed and dominate
the assemblages, which reduces the relative abundance of other mesotrophic taxa (Bárcena et al.,
2004; Hernández-Almeida et al., 2011). Then, multiple recurring high productivity events occurring
over time in the Sicily Channel could explain the amount of *G. bulloides* in both the MARGO core-
tops and the sites 342 and 407. In that sense, the recent warming and stratification of the
Mediterranean could explain the recent trend in the planktonic foraminifera population registered
by the sediment trap. However, in that case, species such as *G. ruber* and other oligotrophic species
should be at least as much represented as in the seabed sediment. Alternatively, this could imply a
change in the intensity of the water masses flowing, such as an increased mesotrophic MAW
influence and a reduced oligotrophic LIW influence.
Additionally, from a chronological point of view, we propose that the main assemblage change
between the settling and the seabed sediment assemblages (i.e. the dominance of *G. inflata*) took
place during the late Holocene but preceded the industrial period. The Incarbona et al., (2019) dates
showed that, overall, since 1558 CE, *G. inflata* already dominated the samples. Also, the chronology
in the work from Margaritelli, (2020) coupled with the abundances presented in allowed to show
that, since the Little Ice Age, the three dominant species in the western Sicily Channel are *G. inflata*
followed by *G. ruber* and *G. bulloides*. This brings further confirmation that *G. inflata* dominated the
seabed sediment in the late Holocene, but also to the fact that the shift in the secondary species
(i.e. *G. truncatulinoides* instead of *G. ruber* and *G. bulloides*) is rather recent. Also, we assume that
the discrepancy with the MARGO core-tops sample is the result of the low temporal resolution.

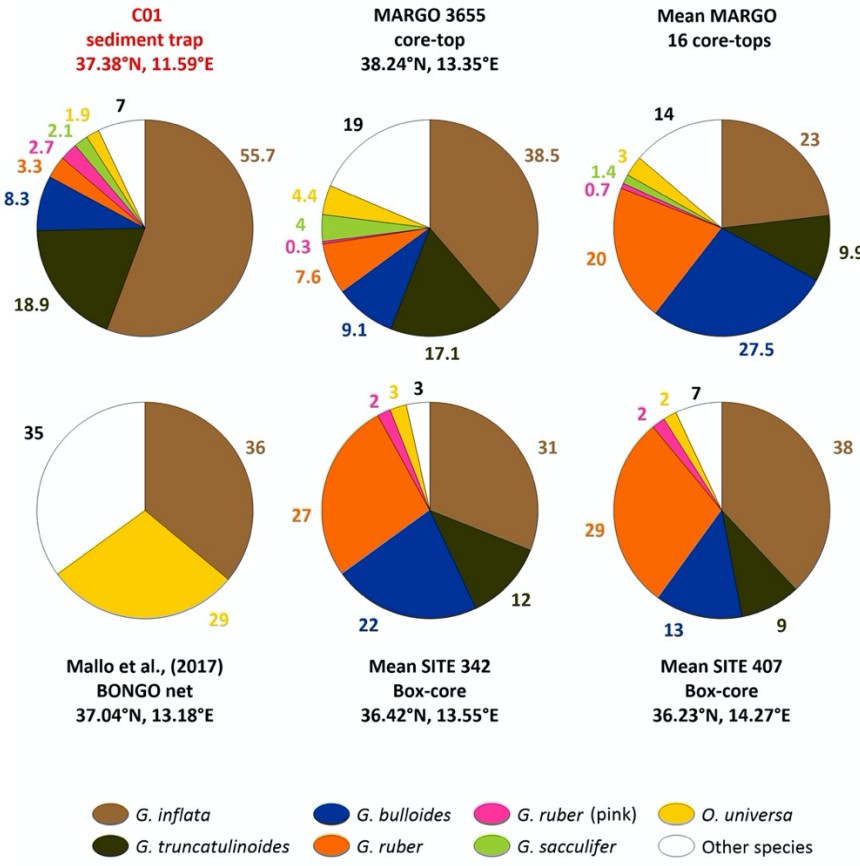


**Figure 8.** Comparison of the relative abundance of the planktonic foraminifera from the sediment trap
and seabed sediment. From top left to bottom right: the settling assemblage from the sediment is
depicted in red; MARGO site 3655 corresponds to the lowest squared chord distance; the mean relative
abundance of all MARGO sites included in this study (see Supplementary data); the results from the
BONGO net retrieved in the Sicily Channel from Mallo et al., (2017); finally, the mean abundances (see
section 3.5) from the two sites presented in Incarbona et al., (2019): sites 342 and 407.

**Table 3.** MARGO core-tops analyzed, their latitude and longitude and the squared chord distance (SCD)
between the sediment trap in the C01 mooring line and the MARGO database core-tops. The complete
SCD for all sites can be found in Supplementary data.

| Site | MARGO database | | | | | | | | | | | | | | | |
|---|---|---|---|---|---|---|---|---|---|---|---|---|---|---|---|---|
| | 3655 | 3677 | 3724 | 3739 | 3737 | 3738 | 3658 | 3725 | 3654 | 3680 | 3735 | 3736 | 3673 | 3727 | 3661 | 3726 |
| Latitude | 38.25 | 36.47 | 35.85 | 36.73 | 38.33 | 38.00 | 36.68 | 36.49 | 38.22 | 37.46 | 38.17 | 38.23 | 39.40 | 38.93 | 39.41 | 38.64 |
| Longitude | 13.35 | 11.49 | 13.03 | 13.95 | 11.80 | 11.78 | 12.28 | 13.32 | 13.27 | 11.55 | 11.23 | 11.25 | 13.34 | 10.59 | 13.34 | 10.78 |
| SCD to C01 | 0.27 | 0.52 | 0.55 | 0.56 | 0.66 | 0.78 | 0.84 | 0.85 | 0.88 | 0.89 | 0.90 | 0.93 | 1.03 | 1.03 | 1.07 | 1.10 |


To document the differences between the assemblage in the C01 mooring line and the MARGO database core-tops, we hereby analyze the SCD between the annual integrated settling foraminifera assemblage of the C01 mooring line and all the core-tops located in the Sicily Channel (see Supplementary Figure 2). Overall, the SCD ranged between 0.27 and 1.1 (Table 3). By using a dissimilarity coefficient value of <0.25 as cutoff criteria (see section 3.6 for more details), it can be concluded that none of the core-tops assemblages can be considered close analogues to the C01 mooring line. The only exception might be MARGO site 3655, located around 180 km northeast of the mooring line, which displayed an SCD value of 0.27, very close to our cutoff threshold. Interestingly, from a geographical point of view, the geographical closest site analyzed (MARGO 3680) displayed a high SCD (0.89) despite being retrieved virtually in the underlying sediments beneath the C01 mooring line (Table 3). Overall, the 4 most similar sites (SCD <0.6) to the settling assemblage are all located eastward, while the 4 most different sites (SCD >1) are all located northward to the location of the mooring line. This highlights the geographical variability of the Sicily Channel regarding the planktonic foraminifera population and the complex oceanographic conditions. Note that, as mentioned previously, the lack of dating in these samples do not allow to bring further interpretations about the timing of planktonic foraminifera populations shifts. In addition to the lack of chronology control in these samples, no data is available for the sedimentation rate, which makes any assumption around the intensity of the hydrodynamics impossible. Finally, and as mentioned earlier, the retrieval method applied for the different core-tops could also be cited as source of the differences between the MARGO core-tops and with the sediment trap in the C01 mooring line. While a box-corer was used for sampling in sites 342 and 407 (Incarbona et al., 2019), various devices were used for the MARGO core-tops, that includes piston and gravity cores that are known to often experience stretching or loss of material during the recovery of the sediments. Therefore, it is likely that the different MARGO surface sediment data set represent different time intervals.

Taken into consideration all the uncertainties presented above, our data suggest that a change in the composition of the planktonic foraminifera assemblages took place at some stage of the late Holocene but before the onset of the industrial period. However, the available data precludes the determination of the main environmental drivers causing this change.

**Conclusions**

The C01 mooring line, located on the axis of the Sicily Channel, provided the opportunity to document the planktonic foraminifera population on an interannual scale. We analyzed 19 samples that covered the timespan between November 2013 and October 2014. A total of 3723 individuals and 15 different species were identified. *G. inflata*, *G. truncatulinoides*, *G. bulloides*, *G. ruber* and *G. ruber* (pink) were the five most abundant species, representing 56, 19, 8, 3.5 and 3% of the total foraminifera. The remaining species represented less than 5% of the total individuals. Total planktonic foraminifera flux ranged between 44 and 1890 shells m$^{-2}$ d$^{-1}$, higher values were reached during spring while values were lower during summer. Our data indicates that the planktonic foraminifera fluxes mainly reflect the oceanographic configuration of the Sicily Channel and its seasonal surface circulation variability. During winter and spring, a stronger eastward advection

favours the MAW entrance in the Sicily Channel, allowing cool and nutrient enriched waters to enter
the Channel. This resulted in an increased planktonic foraminifera flux and a higher presence of *G.*
*inflata*, *G. truncatulinoides* or *G. bulloides*, which are taxa associated with the western basin. On the
other hand, during summer, the eastward advection is reduced and the LIW dominates the water
column, favorizing the increase of species associated with the eastern basin, such as *G. ruber*, and
*G. ruber* (pink). Our correlation data with both SST and chlorophyll-*a* showed that *G. inflata* was
associated with cool and nutrient rich waters. In contrast, both *G. ruber* species were associated
with warm and oligotrophic waters, which agrees with their ecology. Surprisingly, no significant
trends were identified for either *G. truncatulinoides* or *G. bulloides*. As *G. bulloides* flux increased
coincidently with the benthic foraminifera one, we considered that this species might have a
resuspended origin. The comparison with integrated annual data from other sediment trap
experiments conducted in in different regions of the Mediterranean basin, our fluxes and diversity
data indicated that the Sicily Channel can be considered a transitional zone in regard to planktonic
foraminifera populations: annualized fluxes were lower compared to the westernmost Alboran Sea,
but higher than in the easternmost Levantine basin. However, the Sicily Channel exhibited the
highest diversity values across all the sites analyzed, highlighting the influence of both the western
and eastern basins. Finally, the planktonic foraminifera assemblages from the sediment trap were
also compared with seabed sediment assemblages. Overall, both eutrophic and oligotrophic taxa
were more abundant in the seabed sediment, however, *G. inflata* dominated the assemblages in
the closest samples to the sediment trap location. Our dataset was similar to the assemblages from
sites 342 and 407 (Incarbona et al., 2019) but different than the ones from the MARGO core-tops.
This is likely due to the fact that they represented different time periods. Finally, the high-resolution
chronology from sites 342 and 407 allowed to show that the planktonic foraminifera population
shift likely developed during the late Holocene prior to the industrial period. However, the causes
of this shift remain uncertain, and our results call for increasing the monitoring of planktonic
foraminifera populations and accentuating the comparisons between recent and seabed sediment
assemblages in the Mediterranean to determine if the trends suggested by our data are the result
of the recent environmental change.
*Data availability.* All data used in this study are presented in the Supplement and are available online
at doi: 10.17632/tp4v6hm7dc.1 (Béjard et al., 2023).
*Supplement.* The supplement related to this article is available online at:
*Author contributions.* ASRH, FJS and TMB designed the study. JPT designed Fig. 1 and contributed to
planktonic foraminifera identification and imaging. ASV and ILC provided the JERICO C01 sediment
trap samples and led the sample processing. TMB led the microscopy and image analysis, the
foraminifera study, statistical analysis and wrote the manuscript with feedback from all authors.
*Competing interests.* The contact author has declared that none of the authors has any competing
interests.

*Acknowledgements.* The authors would like to thank Aidan Hunter from the BAS (Cambridge) for
the statistical analysis inputs and Francesca Bulian (University of Groningen) for benthic
foraminifera identification support.
*Financial support.* This research has been supported by the Ministerio de Ciencia e Innovación (grant
nos. RTI2018-099489-B-100, PID2021-128322NB-100, and PRE2019-089091). This research has also
received funding from the JERICO project under the FP7 contract agreement nº 262584 and
supported by ISMAR, CNR. ASV acknowledges the financial support by the Catalan Government
Grups de Recerca Consolidats grant (2021 SGR 01195). This project has received funding from the
project BASELINE (grant no. PID2021- 126495NB-741 C33) granted by the Spanish Ministry of
Science and Innovation and Universities (Andrés S. Rigual-Hernández).

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
