# Peer review of "Planktonic foraminifera assemblage composition and flux dynamics inferred from an annual"

_EGUsphere, 2023_

## Author Comment (AC1)

First of all, the authors would like to thank reviewer 1 for the input and all the ideas suggested. We think that the comments made helped to significantly improve the manuscript. In the manuscript, find the changes suggested by reviewer 1 depicted in red. Here, to be clear and precise with our answers, we used **R#-C#** and bold notation for reviewer 1 comments, and our answers appear as **R#-C#.**

**R1-C1: While I acknowledge the authors' characterization of the Mediterranean as a 'miniature ocean' and recognize the significance of publishing foraminifera datasets from this relatively understudied sea, I maintain skepticism regarding the reliability of the results as an ecological signal. This skepticism arises primarily due to the presence of benthic specimens in the sediment trap, which allows for the interpretation of results influenced by hydrodynamics. In essence, the results represent a mixed signal of both ecology and hydrodynamics. For instance, the increased occurrence of deep-dwelling specimens may be attributed to the preferential settling of heavier specimens and the winnowing of lighter ones at the trap site. Similarly, the resemblance of the assemblage to that of the eastward core top sample could be a result of sediment winnowing from the predominantly westward flowing LIW water mass (200 m to 400 m) at the intermediate trap collection depth.**

**R1-R1:** Authors appreciate the point raised by reviewer 1. We agree with reviewer 1 in that the presence of benthic foraminifera raises the question of the role played by the hydrodynamics in the materials collected by the trap. There are several lines of evidence that strongly suggest that the foraminifera flux seasonality collected by the trap is mainly driven by changes in the production and export of foraminifera in the upper water column. Firstly, the seasonal cycle of planktonic production and export collected by the trap is consistent with many other settings within the Mediterranean and the world's oceans (Avnaim-Katav et al., 2020; Bárcena et al., 2004; Rigual-Hernández et al., 2012), displaying maximum values during winter and spring, thereby coinciding with the productive period. Secondly, the composition of the sinking foraminifer assemblages collected by the trap is in good agreement with the composition of the living foraminifera populations dwelling in the overlying water masses (Mallo et al., 2017). The latter used BONGO nets to analyze the planktonic foraminifera population during spring in a pan-mediterranean approach and documented that *G. inflata* was also the dominant species at the moment of sampling (during spring) alongside with *O. universa*. Please note, that this latter argument was not stated in the submitted version of the manuscript, but it is now in the updated version of the manuscript, **chapter 5.5, lines 919-921**. Lastly, it should be noted that although our data indicates that the fluxes collected by the trap are mainly a controlled by pelagic sedimentation, there is clear evidence of influence of resuspended materials into the trap as suggested by the presence of benthic foraminifera. However, we acknowledge that benthic foraminifera are registered at low numbers during the whole record (on average 3.3%), and this is the reason why background influence of resuspended materials was indicated in the manuscript (**lines 474-475** of the first version of the manuscript). In terms of contribution to the annualized foraminifera flux, benthic foraminifera only represented 1.1% of the total flux. Of those 1.1%, 80% was collected during the month of April 2014.

However, we acknowledge that not all the arguments highlighted above were clearly explained in the first version of the manuscript and therefore, they will in the corrected version of the manuscript. Information about benthic foraminifera is now more complete at **chapters 4.1., 4.3 and 5.1., lines 358-360 , 402-404 and 514-525 respectively.** Note that more information about the benthics individuals was also requested by reviewer 1 and therefore appears

As an comparison, the Planier sediment trap, located in the axis of the Planier canyon in the Gulf of Lions and located 500 m above the seafloor registered 3.5% of benthic individuals with relative abundances ranging between 0.5-11% (unpublished data). In the Alboran Sea, the ALB5F sediment trap registered 2.9% of "benthic-neritic" (Bárcena et al., 2004). Our findings about benthic individuals are within the range of the previous work with which we compare our data. However, we acknowledge that the hydrodynamics are likely playing a role during the spring period.

Concerning the occurrence of deep dwellers, we acknowledge that the point raised by reviewer 1 is a possibility. In addition, as reviewer 2 suggested, we added a comparison with the work of Mallo et al., (2017). The latter used BONGO nets to analyze the planktonic foraminifera population during spring in a pan-mediterranean approach. The site located in the Sicilian Channel showed that, in surface waters, *G. inflata* dominated the assemblage, alongside with *O. universa*. Therefore, our most abundant species is in accordance with the most abundant from the surface record, which, in our opinion, rules out the resuspended origin of this taxa. The questions now transfers to the secondary taxa. *O. universa* seems to be particularly high alongside the Algerian coast (Azibeiro et al., 2023), however, the machanisms behind its distribution in the Mediterranean remain poorly constrained (Mallo et al., 2017).

Finally, concerning the LIW influence as a possible explanation for the resemblance of our assemblages with the eastern core-tops, the authors agree. Once again, reviewer 2 provided a useful input, which is to use the Incarbona et al., (2019) box-core samples to have a better constrained chronology. These samples were also retrieved in the Sicily Channel and appeared to show more similarities with our assemblage. Specially, again, concerning *G. inflata* as the main taxa. So now, *G. inflata* is the dominant taxa in the surface, intermediate depth (our assemblage) and the seabed sediment. However, this samples showed a high proportion of *G. ruber* (white). In turn, this species appears in a much reduced proportion in our samples (see next comment for the shell weight discussion).

**R1-C2: More specifically, it has been shown that planktonic foraminifera calcify in accordance with their habitat depth, with species dwelling at deeper levels producing heavier shells compared to those inhabiting surface waters (Zarkogiannis et al., 2022). For instance, *G. truncatulinoides* typically generates among the heaviest shells (see previous), as does *G. inflata* when compared to *G. ruber* and *G. bulloides* (Feldmeijer et., 2013). Consequently, certain current speeds may favor the settling of specific species specimens while others are dispersed elsewhere, potentially explaining the observed counts. In the**

**central Mediterranean,** *G. bulloides* **is more prevalent in sediments (and thus resuspended sediment) than** *G ruber***, contributing to a simultaneous peak in benthic foraminifera counts. Hydrodynamics may therefore account for discrepancies in cases where specific environmental drivers cannot be identified for certain species, leading to the invocation of other environmental controls in the text to explain the observed patterns. Indeed hydrodynamics in the area are particularly strong especially in spring (Gasparini et al., 2004), while sediment resuspension in the wider area is found to peak in spring (Grifoll et al., 2019) as well. Data from Copernicus also indicate increased flow speeds at 400m during spring of 2014.**

R1-R2: Authors agree that generally, the deep-dwelling species such as *G. truncatulinoides* and *G. inflata* are among the heaviest planktonic foraminifera species. Particularly compared to surface dwellers (Beer et al., 2010; Béjard et al., 2023). However, we find unlikely that differences in foraminifera shell weights could be responsible for the differences between the sediment trap and surface sediments. Firstly, and as mentioned in the previous comment, *G. inflata* is the most abundant species in both the surface BONGO nets, in our sediment trap, and in the seabed sediment (with [210]Pb chronology as a support, see Incarbona et al., (2019)). The abundances between these datasets are similar, highlighting a dominance of this species in the modern central Mediterranean Sea. Secondly, under the winnowing theory, *G. bulloides*, a lighter species very abundant in the sediment, should travel very far with strong currents and be very abundant in the surface waters , however, it is under represented in our sediment trap and even absent from surface BONGO nets collected during spring (Mallo et al., 2017). Thirdly, *G. ruber*, which is lighter than the deep-dwelling species, but heavier than some surface dwellers such as *G. bulloides* (Beer et al., 2010; Weinkauf et al., 2016). Interestingly, *G. bulloides* is almost twice more abundant in our samples than *G. ruber*, under the winnowing theory, the opposite trend should be expected. Finally, it could be argued that *G. truncatulinoides*, as the heaviest species, should dominate the sediment trap samples, however, it is less abundant than *G. inflata*, which dominates the seabed sediment. In a winnowing theory, *G. truncatulinoides* should show a higher abundance in the seabed.

Finally, we also analyzed the data suggested by reviewer 1, from Copernicus, that states that the flow speed increased during spring 2014. The data we retrieved from 400m deep (mean Mediterranean sea water velocity) showed that the flow increase happened during February 2014. The flow speed for April 2014 is rather low. Both the monthly and daily data showed that the peak of current flow developed during winter. Which on the other hand also contradicts previous work such as Gasparini et al., (2004).

However, as a conclusion of the previous two comments, we agree with reviewer 2 that, during spring, the current speed increase. Then we also agree the hydrodynamics might be playing a role and affect the distribution of taxa such as *G. bulloides* and *G. truncatulinoides*. Therefore, we added a new discussion chapter: **chapter 5.3**, named **"Influence of the hydrodynamic conditions on the planktonic foraminifera assemblage"**. It consists mainly of a discussion around the impact of hydrodynamics with all the bibliographic input

provided by reviewer 2. In the end chapter we disclose the possibility of the winnowing impact during spring and overall, during higher current settings. In that regard, the **introduction** and **conclusion** have also been updated accordingly. **Lines 43-45 and 1058-1060**, respectively.

**R1-C3: MARGO site areas should be checked for sedimentation rates, as regions with high sedimentation will likely experience reduced hydrodynamics, facilitating the settling of lighter, surface-dwelling specimens. Additionally, for any inferences regarding ecosystem shifts in the Mediterranean, the sediment traps in other locations should be compared with nearby sedimentary material. Certainly, a pan-Mediterranean comparison should evaluate whether the data from the current study from the Sicily strait (area of high velocities) should be considered or disregarded.**

**R1-R3:** Authors agree with the suggestion that the sedimentation rates should be checked as they could provide useful information about the hydrodynamic context. Unfortunately, 9 of the retrieved core-tops appear as "unpublished", while the publications belonging to the remaining material do not provide the sedimentation rate (Thunell, 1978). Therefore, to provide a more complete picture of the seabed sediment assemblages, now the MARGO sites are not the only seabed sediment used, now the Incarbona et al., (2019) also appear in the manuscript. In the latter work, sites 342 and 407, located in the Sicily Channel slightly eastward of the C01 sediment trap, were analyzed and compared with water samples collected in the close vicinity of the C01 sediment trap. Additionally, and as mentioned earlier, the chronology is based on [210]Pb. In these samples, *G. inflata* also dominates the assemblages, followed by *G. ruber* and a similar proportion of *G. bulloides* as the one found in our samples. The MARGO samples are still discussed in **chapter 5.5**, but in much a reduced way.

Concerning the pan-mediterranean comparison, we did not include samples covering the whole Mediterranean basin because the novelty of our work is mainly the planktonic foraminifera data from the C01 sediment trap. In that regard, we wanted, firstly, to put this assemblage in perspective with other mooring lines (**chapter 5.4**) and then compare it with the seabed sediment in the central Mediterranean to possibly identify assemblages changes during recent times. There are various reasons why we limited our sediment comparison within the Sicily Channel. The distance between the C01 sediment trap and the seabed sediment is around 243km to the westernmost core-top (MARGO 3727) and 213km to the easternmost core-top (MARGO 3724), which in turn shows that the area covered is considerable. Also, we only included core-tops located in a 2.5 degres distance in order to only display potentially comparable seabed assemblages to the sediment trap. Finally, the concept of pan-mediterranean approach was not intended in our original version of the manuscript. Rather a "put into perspective" strategy.

However, as the comparison with the seabed sediment raised questions, we re-designed **chapter 5.5**. We now discuss the similarities and differences between the C01 sediment trap and the different seabed sediment datasets (MARGO and sites 342 and 407). We

discuss the possibility of the retrieval device (i.e. core-top, box-core) impact on the sediment preservation. We now also discuss the similarities between our assemblages and the surface ones (Mallo et al., 2017). We also acknowledge the lack of sedimentation rates and the winnowing and the sediment resuspension as a possible explanation of the recent assemblages for the MARGO database, **lines 1012-1014**. The sum of the reasons we display within our comparison allow us to document a change in the planktonic foraminifera population during the Holocene, and. Although we propose the Incarbona et al., (2019) chronology, we cannot state with precision the exact timing of the latter. Note that the **introduction** and **conclusions** have been modified accordingly. **Lines 53-58 and 1066-1073.**

**R1-C4: Furthermore, in a future submission please change planktic to planktonic. The correct adjective form of plankton in Greek is planktonic. The adjectives of Greek nouns ending in -on get the suffix -ic in the end like plankton – planktonic, bion – bionic, lacon – laconic (preserved also in French words like Napoleon – Napoleonic). This is different to nouns ending in -os, which lose the ending -os to the previous consonant by replacing it with -ic, like bentos – benthic, cosmos – cosmic or chronos – chronic.**

**R1-R4:** Authors agree and understand the linguistic justification behind this comment. The term "planktic" has been replaced by "**planktonic**" in the whole manuscript. Note that not all the planktonic terms have been depicted in red, just a couple of examples.

**References**

Bárcena, M. A., Flores, J. A., Sierro, F. J., Pérez-Folgado, M., Fabres, J., Calafat, A., and Canals, M.: Planktonic response to main oceanographic changes in the Alboran Sea (Western Mediterranean) as documented in sediment traps and surface sediments, Marine Micropaleontology, 53, 423–445, https://doi.org/10.1016/j.marmicro.2004.09.009, 2004.

Beer, C. J., Schiebel, R., and Wilson, P. A.: Technical Note: On methodologies for determining the size-normalised weight of planktic foraminifera, Biogeosciences, 7, 2193–2198, https://doi.org/10.5194/bg-7-2193-2010, 2010.
Béjard, T. M., Rigual-Hernández, A. S., Flores, J. A., Tarruella, J. P., Durrieu De Madron, X., Cacho, I., Haghipour, N., Hunter, A., and Sierro, F. J.: Calcification response of planktic foraminifera to environmental change in the western Mediterranean Sea during the industrial era, Biogeosciences, 20, 1505–1528, https://doi.org/10.5194/bg-20-1505-2023, 2023.
Incarbona, A., Jonkers, L., Ferraro, S., Sprovieri, R., and Tranchida, G.: Sea Surface Temperatures and Paleoenvironmental Variability in the Central Mediterranean During Historical Times Reconstructed Using Planktonic Foraminifera, Paleoceanog and Paleoclimatol, 34, 394–408, https://doi.org/10.1029/2018PA003529, 2019.
Mallo, M., Ziveri, P., Mortyn, P. G., Schiebel, R., and Grelaud, M.: Low planktic foraminiferal diversity and abundance observed in a spring 2013 west–east Mediterranean Sea plankton tow transect, Biogeosciences, 14, 2245–2266, https://doi.org/10.5194/bg-14-2245-2017, 2017.

Rigual-Hernández, A. S., Sierro, F. J., Bárcena, M. A., Flores, J. A., and Heussner, S.: Seasonal and interannual changes of planktic foraminiferal fluxes in the Gulf of Lions (NW Mediterranean) and their implications for paleoceanographic studies: Two 12-year sediment trap records, Deep Sea Research Part I: Oceanographic Research Papers, 66, 26–40, https://doi.org/10.1016/j.dsr.2012.03.011, 2012.

Thunell, Robert C: Distribution of planktonic foraminifera in surface sediments of the Mediterranean Sea, https://doi.org/10.1594/PANGAEA.55624, 1978.

Weinkauf, M. F. G., Kunze, J. G., Waniek, J. J., and Kučera, M.: Seasonal Variation in Shell Calcification of Planktonic Foraminifera in the NE Atlantic Reveals Species-Specific Response to Temperature, Productivity, and Optimum Growth Conditions, PLoS ONE, 11, e0148363, https://doi.org/10.1371/journal.pone.0148363, 2016.

---

## Author Comment (AC2)

First of all, the authors would like to thank reviewer 2 for such a precise and helpful review of our work. We think that the suggestions and comments made helped improve the manuscript. In addition, the questions raised also made us re-think a couple of the sections. In the manuscript, find the changes suggested by reviewer 2 depicted in blue.

Here, to be clear and precise with our answers, we used **R#-C#** and bold notation for reviewer 2 comments, and our answers appear as **R#-R#.**

**R2-C1: In terms of missing references:**

**The sediment trap site is located in the same area analyzed by K. Schroeder, J. Chiggiato, S. A. Josey, M. Borghini, S. Aracri & S. Sparnocchia (2017). Rapid response to climate change in a marginal sea.** *Scientific Reports* **| 7: 4065 | DOI:10.1038/s41598-017-04455-5. It is very strange that this reference is not reported in the submitted manuscript.**

**In addition, concerning the oceanography of the Mediterranean is not reported as reference Pinardi et al., (2015). Mediterranean Sea large-scale low-frequency ocean variability and water mass formation rates from 1987 to 2007: A retrospective analysis.** *Progress in Oceanography* **132 (2015) 318–332.**

**and in the following paper (Garcia-Solsona et al., 2020) one of the study station is very close to the sediment trap location. Maybe you could find additional information for the submitted manuscript.**

**Garcia-Solsona E., Pena L. D., Paredes E., Pérez-Asensio J.N., Quirós-Collazos L., Lirer F., Cacho I., (2020). Rare Earth Elements and Nd isotopes as tracers of modern ocean circulation in the central Mediterranean Sea. Progress in Oceanography, 185. https://doi.org/10.1016/j.pocean.2020.102340 (7) -**

**R2-R1:** Authors appreciate the references suggested by reviewer 2. They are now included in the manuscript in **lines 152-630-708-709** (as an example).

**R2-C2:** *Chapter 3.3:*

**The authors report line 218 that: G. bulloides includes G. falconensis [(this is my interpretation when I see in the text Globigerina bulloides (+ G. falconensis)]. Maybe my interpretation is wrong. But, if it is right my interpretation, the question is that in the text (Table 1, or line 363) the authors consider separately these two species. Please try to control, if necessary.**

**In the plate is missing the picture of Turborotalita quinqueloba. Just a question: did you find only Globigerinoides with chambers lobulate (trilobus) or also with elongate and sac-like (sacculifer), and if you found both, what is the abundances of these morphotypes? Are the present in the same seasons?**

**R2-R2:** Authors agree with this remark, it is odd. As only one individual of *G. falconensis* has been identified through the samples, we decided to remove it from Table 1 and specifiy in a clearer way that only one of this taxa has been counted in **Lines 224-225** "Finally, only one individual of *G. falconensis* has been identified**."** Also, now the **Table 1** caption states : "Note that *G. falconensis* has not been included due to its scarcity (only one individual was identified)."

At first, we did not include a picture of *T. quinqueloba* to **Plate 1** because our intention was to showcase only the most abundant and representative species. However, in the new version of the manuscript, **this species has been included**.

Concerning the differentiation between lobulated and sac-type *Globigerinoides*, we mainly found individuals belonging to the first group, the sac-type individuals were scarce. The latter were identified mainly during summer and autumn. This also appears in **chapter 4.1, lines 355-357.**

**R2-C3: *Chapter 3.4:***

**Line 238-239: it means that the SST data in the same site of sediment trap are also from mooring line data. Is it wright? In several figures of the submitted manuscript, are from satellite derived or from data in situ the SST data? In is not clear for me.**

R2-R3: Authors apologize for this confusion. No data comes directly from the mooring line, as reviewer 2 states later in the comment, all the SST and chlorophyll-a data represented in the figures are satellite-derived. We simplified the first sentence of the chapter in order to amplify the satellite side of it. **Line 247** "To assess the possible relationship of planktic foraminifera fluxes with environmental variability,  satellite-derived chlorophyll-*a* and Sea Surface Temperatures (SSTs) were retrieved from global data sets." Furthermore, the section has been divided into two new sections: **3.4 Satellite-derived environmental parameters** and **3.5. Planktonic foraminifera flux and surface sediment data from other Mediterranean settings**.

**R2-C4: *Chapter 4.3:***

**Please in Fig. 3, report the planktonic foraminiferal species according to the % abundances: 1) G. inflata, 2) G. truncatulinoides, 3) G. bulloides, 4) G. ruber pink, 5) G. ruber white, 6) O. universa, 7) G. rubescens…….**

**I do not understand why the authors decided to plot in Fig. 3 the planktonic foraminiferal data also in %. I think that the authors have to plot only the flux (shells). In many cases the strong differences in shells, between the different months, produced an altered % signatures. The impact on % abundances it is possible to observe on G. truncatulinoides signal. According to the ecological niche of G. truncatulinoides is an indicative species of deep vertical mixing during the winter season, so that it is very strange to observe high abundance % of this species during the summer season of sediment trap data. Is there an explanation of this discrepancy? In fact, if you consider only the flux signals, G. inflata and G. truncatulinoides are in phase according to their ecological preferences.**

**Moreover, the Gulf of Lion sediment trap data, during winter season, document that the high abundance of G. truncatulinoides results almost in phase with high abundance of G. bulloides. And it has sense. In your record, this relation is not evident. Is there is an explanation for this discrepancy? The ascended of G. truncatulinolides to the euphotic zone, where it proliferates due to strong advection of nutrients from the nutrient-rich deeper layers and consequently high primary productivity could be supported by the increase in abundance of nutrient rich species G. bulloides. In the Sicily Channel this**

relation seems not documented, probably related to the occurrence of other oceanographic influence. Have you an idea?

Can the authors explain the criteria adopted for the season's boundaries? I think that the spring season has to start with sample 5-april and not before and also 1-june is summer and not Spring.

**R2-R4:** Authors agree that the relative abundances signal (in %) can sometimes be altered by a low number of individuals in the samples and therefore produce "extreme" results and lead to various interpretations. Therefore, now **Figure 3** only shows the fluxes and the species are plotted according to their abundance order, as suggested. Furthermore, the same figure with the relative abundances has been added to supplementary material (Supplementary Figure 3).

Consequently, the results **chapter 4.3**. has been modified to be more direct and spend less to the explanation of the seasonal abundances. **Lines 411-421.**

Concerning the questions asked, in our opinion, the discrepancy between the timing and abundance of *G. bulloides* and *G. truncatulinoides* resided in the amount of nutrients and the productivity of the water masses. In the Gulf of Lions, *G. bulloides* is the main species and shows the classical "bloom" behaviour, while *G. truncatulinoides* is present more constantly and its variations are more gradual. Although the timing of the two species is different here, the response of *G. truncatulinoides* is similar across the record. Furthermore, from a productivity standpoint, the Sicily Channel is less productive than the Gulf of Lions, which in turn does not benefit *G. bulloides* abundances and, as the upwelling in our study zone is less pronounced, the timing between the two species is different. Furthermore, the upwelling conditions in the central Mediterranean are caused by the LIW flowing to the western part of the basin, which leads to reduced productivity upwelling conditions. This could explain the lack of *G. bulloides* blooms here. Therefore, we agree with the suggestion made by reviewer 2 that other oceanographic processes must be playing a major role here. This discussion is now included in the new version of the manuscript in **chapter 5.2. lines 616-633.**

Finally, the seasons have been established according to the astronomical sense of the term (i.e. equinoxes and solstices). They have not been considered within the samples (i.e., spring begins with the first sample considered to be collected during spring). As we are in a temperate zone of the ocean, with a temperate climate and we are located in the northern hemisphere, we therefore consider Winter from the end of December to the end of March, Spring from the end of March to the end of June, Summer from the end of June to the end of September and Autumn from the end of September to the end of December. This was also made to be coherent with the other datasets with which we compared our data. However, if reviewer 2 does not agree, we'll change the season's settings. The seasons description now appears in **chapter 3.3 lines 242-244: "Lastly, to describe the seasonal flux variations and to put our results into a regional context and be coherent with previous studies, each season was defined as spring (March–May), summer (June–August), autumn (September–November) and winter (December–February)."**

**R2-C5:** *Chapter 5.1:*

I am very curious if in terms of benthic foraminifera, the authors found only these two species or is there a diversified benthic assemblage? If they found a diversified benthic assemblage, the identifies species are related to the same environment. Did you consider this issue?

In addition, the authors considered the benthic species a result of resuspended sediments process. The question is as follows: In the seasons where you find benthic species, did you also find altered assemblages in planktonic foraminifera (also change in size and/or not well preserved planktonic foraminifera)? I think that this is important to verify the reliability of the samples containing benthic foraminifera. The sediment trap is located ca. 400 meter deep and I can image that the suspended foraminifera could be related to strong LIW activity. Is it possible?

**R2-R5:** Reviewer 2 asks an interesting question. When we decided to discuss the presence of benthic foraminifera as a possible indicator of resuspended material, we focused on the population variability. Overall, the two species described in the manuscript dominate the benthic assemblage. We also identified some *Uvigerina mediterranea* and *Lagenidae* type individuals (infaunal and epifaunal genus and species), although the number of the latter were much lower. This is also specified in **lines 509-511.** Overall, benthic foraminifera mainly appeared during early April and early June, specifically in 4 samples (see Supplementary material), overall, around 3723 planktonic foraminifera individuals were identified against only 141 benthic individuals.

Concerning the preservation state of the planktonic foraminifera in the samples that contain a relatively high amount of benthic individuals, they were well preserved, in the same state as the remaining samples. Despite not having measured the dimensions of the foraminifera, visually, the deep dwellers (*G. inflata* and *G. truncatulinoides*) seemed a little bigger. However, in those samples, we identified a higher amount of detritic material: mica flakes and framboidal pyrite. Those samples also contained a higher amount of *G. bulloides*, which is supposed to be the dominant species in the seabed sediment and to come from the MAW influence. As the questions mentions, the influence of the LIW during this period, which starts to affect the Channel in April but grows in intensity during summer, could be a factor affecting the distribution of benthic foraminifera. However, as the amount of benthic foraminifera presents some variability and is maximum in three samples that do not belong to the period of maximum LIW intensity (summer to early autumn) and neither to the maximum intensity of the MAW (winter), we do not consider their presence nor relative abundance as a reliable proxy for MAW/LIW intensity. We then interpret this increase of benthic foraminifera as a punctual increase of the current speed in the Sicily Channel. We included this discussion in **chapter 5.1. lines: 515-525**.

**R2-C6:** *Chapter 5.4:*
I have several doubts concerning the possibility to compare the sediment trap data with information from coretops from MARGO database. The age of these coretops is strongly different from the analysed sediment trap short time series. In fact, the authors reported that in seabed sediments *Globigerina bulloides* represents the main species.
Data on planktonic foraminifera (Margaritelli PhD thesis, 2016, Perugia, Italy) from a special gravity core system SW104, that allows the recovery of undisturbed and wellpreserved water–sediment interface, show that the main species over the last 100 century (according to radionuclides chronology published in Margaritelli et al. 2020) is first *Globorotalia inflata* followed by *Glibogerinoides ruber* white variety, while *G. bulloides* represents the third abundance species. This discrepancy with coretops MARGO data is mainly related to the low resolution chronology of the coretops due to the missing of radionuclides ages (in my opinion). It means that you compare present day data with a mean signal over the latest part of the Holocene.

Incarbona et al 2019, analysed the planktonic foraminifera over the last four centuries and it is evident that the Sicily Channel is a complex system (from west to the east part). Anyway, the analyzed site (Site 342) in Incarbona et al. (2019) shows over the last century, high abundance values of G. ruber, G. bulloides and last G. inflata, conversely, Site 407 shows high abundance of G. inflata, G. ruber and last G. bulloides. The chronology of these sites is based on radionuclides ages and these data seem to support that the comparison with MARGO database could be questionable. Did you consider also the sites published in Incarbona et al 2019 (Paleoceanography and Paleoclimatology https://doi.org/10.1029/2018PA003529)?

I am reporting these data to stress the fact that the coretops database is useful to reconstruct glacial/interglacial sea surface temperature but, in my opinion, it is very difficult to use this database when comparing with sediment trap record. If the coretops data have a strong chronological control, obviously the comparison could be possible. Is the chronology of the MARGO coretops based con radionuclide ages?

The proposed interpretation concerning the discrepancy between sediment trap data and MARGO database could work but I am convincing that the comparison is questionable due to the different chronologies of the compared records.

General questions:

In Mallo et al. (2017) the authors reported that the most common size fraction of living planktonic foraminifera from bongo net is 150–350 μm and in the Sicily Channel they found the highest percentages of > 500 μm tests. In the submitted manuscript did you find the same size intervals from sediment trap record? And if no, could you suggest an explanation on this issue?

Concerning the size fraction used for the analysis, >150 micron, useful also related to MARGO database, it is clearly evident that according to this choice you did not consider in planktonic assemblage most of T. quinqueloba, G. glutinata and N. incompta. Did you try to consider the impact of this choice on the planktonic foraminiferal diversity, mainly related to productivity signal?

R2-R6:

Authors acknowledge the lack of dating control on the seabed sediment used in the previous version of the manuscript. Authors also agree with the bibliographic input from reviewer 2, it was very helpful to redesign **chapter 5.5** (ex chapter 5.4) To make the latter more complete, the data from Incarbona et al., (2019) has been added to the discussion and the work from Mallo et al., (2017) has been used as a comparison work for our time-series. Then, **chapters 3.5, 3.6 and 5.5** have been modified accordingly. Overall, we have discussed the differences between the seabed sediments datasets as the result of both oceanographical and environmental changes.

**Figure 7** now features the data from Mallo et al., (2017) and from the two box-core sites studied in Incarbona et al., (2019). Also, one MARGO site was removed from the figure, the most different one.

Also, note that all the discussion around the MARGO database has been reduced and moved at the end of **chapter 5.5., lines 994-1019.** Also, note that **introduction** and **conclusions** have been modified accordingly**, lines 43-45 and 1058-1060.**

Concerning the question about Mallo et al., (2017), specifically, the most abundant size they found was 350-500 micron followed by >500 micron. This is likely due to the fact that the main species they identified were *G. inflata* and *O. universa*, which are considered "big-sized" species. In our samples, the sizes distributions were similar, most of the species were in the vicinity of the 500um. Also likely because *G. inflata* and *G. truncatulinoides* dominated the samples.

Concerning the size selected for the analysis, we choose 150 microns for the sake of comparison between the different sites both in the Mediterranean, but also with other time-series and seabed sediment across the world as it is the most widely used size-fraction for planktonic foraminifera. However, we acknowledge the point raised buy reviewer 2, *T. quinqueloba* and *N. incompta* (amongst others), are small-sized species. According to Chernihovsky et al., (2023), most of the *T. quinqueloba*, *G. tenella* and *G. rubescens* individuals are in the 120-135micron range. As we feel this is an important point, it is now included in **section 3.3, lines 227-231.**

**R2-C7: Please consider to use the term Sicily Channel and not Strait**
**R2-R7:** Authors agree to use the term Channel. The whole manuscript has been updated. Find some examples in **Lines 148-167-525-654.** Note that for readability, not all the "Channels" have been coloured in blue.

**R2-C8: Fig. 5: I think that for the reader it is much clearer if close to the terms Tropical/Subtropical, Temperate/Subtropical and Deep dwellers the authors report also the term group1, 2 and 3, as in the text of the manuscript.**
**R2-R8:** Authors agree. Figure 5 now also displays the groups mentioned in the manuscript.

---

## Author Response (AR1)

Author's response.

Find here a list of the main changes that have been carried out in the manuscript according to Reviewer 1 and Reviewer 2. Note that, compared to the individual reviewer response, some of the parts and lines described previously may have changed, find here the new lines of the main changes corresponding to the **tracked changes version**:

- **Material and methods**. **Section 3.3** has been updated according to Reviewer 2 suggestions surrounding the planktonic foraminifera sizes and seasonality description. Also, according to Reviewer 2, now, **Section 3.4** describes only satellite-derived data while **Section 3.5** focuses on the remaining datasets recovered. The data by Incarbona et al., (2019) suggested by Reviewer 2 appears in **Section 3.5 (Lines 280-289)**. Plate 1 now also features *T. quinqueloba.*

- **Results**. Table 1 does not feature *G. falconensis* anymore. As suggested by Reviewer 2, we added a description of sac-type and non-sac type individuals (**Lines 368-370**). According to Reviewer 1 we added a description of the benthic foraminifera contribution. According to Reviewer 1, **Section 4.3** now features a description of the benthic foraminifera contribution to the recorded foraminifera flux (**Lines 415-416**). As Reviewer 2 suggested, **Figure 3** only shows the species fluxes and, therefore, the description surrounding the relative abundance has been shortened.

- **Discussion.** As suggested by Reviewer 2, in **Section 5.2** a discussion surrounding the interactions between *G. bulloides* and *G. truncatulinoides* has been added (**Lines 618-640**). To answer the suggestion by Reviewer 1, a new section, **Section 5.3**, has been added (**Lines 722-781**). This one focuses on the effects of the hydrodynamics on the planktonic foraminifera fluxes. It also features a new figure, **Figure 6**, which shows the connection between *G. bulloides* and the benthic foraminifera flux, to further document the resuspended origin of the latter species. Also, now this section states in a clearer way the effects of resuspended material on the planktonic foraminifera population. **Section 5.5.** has been reworked. Now, as suggested by Reviewer 2, it features the data from Incarbona et al., (2019) and **Figure 8** shows the relative abundance of the latter work, that include sites 342 and 407.

- **General changes.** As asked by Reviewer 1, now we use the term planktonic instead of planktic. As asked by Reviewer 2, now we use the term Channel instead of Strait. The bibliographic input by both reviewers has also been added to the manuscript.

Find here a list of the changes made by the authors when rewriting some of the manuscript:

- **Introduction, abstract and conclusion**. Modified accordingly to the changes made in the whole manuscript (**Lines 53-57, 129-130 and 1111-1121**).
- **Material and methods**. Sections 3.1 and 3.2 have been updated to clarify the C01 mooring nomenclature (**Lines 196-199 and 212-217**)
- **Results**. Section 4.4, the CCA description has been modified in order to be more precise **(Lines 466-470**).
- **Discussion**. Section 5.2, the *G. ruber* distribution discussion has been modified to clarify its winter distribution (Lines 659-671). Section 5.5, *G. bulloides* distribution in the sediment now features another possible explanation such as a recurrence of high productivity events **(Lines 1001-1008).**

Incarbona, A., Jonkers, L., Ferraro, S., Sprovieri, R., and Tranchida, G.: Sea Surface Temperatures and Paleoenvironmental Variability in the Central Mediterranean During Historical Times Reconstructed Using Planktonic Foraminifera, Paleoceanog and Paleoclimatol, 34, 394–408, https://doi.org/10.1029/2018PA003529, 2019.

**RESPONSE TO REVIEWER 1:**

First of all, the authors would like to thank reviewer 1 for the input and all the ideas suggested. We think that the comments made helped to significantly improve the manuscript. In the manuscript, find the changes suggested by reviewer 1 depicted in red.

Here, to be clear and precise with our answers, we used **R#-C#** and bold notation for reviewer 1 comments, and our answers appear as **R#-C#.**

**R1-C1: While I acknowledge the authors' characterization of the Mediterranean as a 'miniature ocean' and recognize the significance of publishing foraminifera datasets from this relatively understudied sea, I maintain skepticism regarding the reliability of the results as an ecological signal. This skepticism arises primarily due to the presence of benthic specimens in the sediment trap, which allows for the interpretation of results influenced by hydrodynamics. In essence, the results represent a mixed signal of both ecology and hydrodynamics. For instance, the increased occurrence of deep-dwelling specimens may be attributed to the preferential settling of heavier specimens and the winnowing of lighter ones at the trap site. Similarly, the resemblance of the assemblage to that of the eastward core top sample could be a result of sediment winnowing from the predominantly westward flowing LIW water mass (200 m to 400 m) at the intermediate trap collection depth.**

**R1-R1:** Authors appreciate the point raised by reviewer 1. We agree with reviewer 1 in that the presence of benthic foraminifera raises the question of the role played by the hydrodynamics in the materials collected by the trap. There are several lines of evidence that strongly suggest that the foraminifera flux seasonality collected by the trap is mainly driven by changes in the production and export of foraminifera in the upper water column. Firstly, the seasonal cycle of planktonic production and export collected by the trap is consistent with many other settings within the Mediterranean and the world's oceans (Avnaim-Katav et al., 2020; Bárcena et al., 2004; Rigual-Hernández et al., 2012), displaying maximum values during winter and spring, thereby coinciding with the productive period. Secondly, the composition of the sinking foraminifer assemblages collected by the trap is in good agreement with the composition of the living foraminifera populations dwelling in the overlying water masses (Mallo et al., 2017). The latter used BONGO nets to analyze the planktonic foraminifera population during spring in a pan-mediterranean approach and documented that *G. inflata* was also the dominant species at the moment of sampling (during spring) alongside with *O. universa*. Please note, that this latter argument was not stated in the submitted version of the manuscript, but it is now in the updated version of the manuscript, **chapter 5.5, lines**

**919-921**. Lastly, it should be noted that although our data indicates that the fluxes collected by the trap are mainly a controlled by pelagic sedimentation, there is clear evidence of influence of resuspended materials into the trap as suggested by the presence of benthic foraminifera. However, we acknowledge that benthic foraminifera are registered at low numbers during the whole record (on average 3.3%), and this is the reason why background influence of resuspended materials was indicated in the manuscript (**lines 474-475** of the first version of the manuscript). In terms of contribution to the annualized foraminifera flux, benthic foraminifera only represented 1.1% of the total flux. Of those 1.1%, 80% was collected during the month of April 2014. However, we acknowledge that not all the arguments highlighted above were clearly explained in the first version of the manuscript and therefore, they will in the corrected version of the manuscript. Information about benthic foraminifera is now more complete at **chapters 4.1., 4.3 and 5.1., lines 358-360 , 402-404 and 514-525 respectively.** Note that more information about the benthics individuals was also requested by reviewer 1 and therefore appears

As an comparison, the Planier sediment trap, located in the axis of the Planier canyon in the Gulf of Lions and located 500 m above the seafloor registered 3.5% of benthic individuals with relative abundances ranging between 0.5-11% (unpublished data). In the Alboran Sea, the ALB5F sediment trap registered 2.9% of "benthic-neritic" (Bárcena et al., 2004). Our findings about benthic individuals are within the range of the previous work with which we compare our data. However, we acknowledge that the hydrodynamics are likely playing a role during the spring period.

Concerning the occurrence of deep dwellers, we acknowledge that the point raised by reviewer 1 is a possibility. In addition, as reviewer 2 suggested, we added a comparison with the work of Mallo et al., (2017). The latter used BONGO nets to analyze the planktonic foraminifera population during spring in a pan-mediterranean approach. The site located in the Sicilian Channel showed that, in surface waters, *G. inflata* dominated the assemblage, alongside with *O. universa*. Therefore, our most abundant species is in accordance with the most abundant from the surface record, which, in our opinion, rules out the resuspended origin of this taxa. The questions now transfers to the secondary taxa. *O. universa* seems to be particularly high alongside the Algerian coast (Azibeiro et al., 2023), however, the machanisms behind its distribution in the Mediterranean remain poorly constrained (Mallo et al., 2017).

Finally, concerning the LIW influence as a possible explanation for the resemblance of our assemblages with the eastern core-tops, the authors agree. Once again, reviewer 2 provided a useful input, which is to use the Incarbona et al., (2019) box-core samples to have a better constrained chronology. These samples were also retrieved in the Sicily Channel and appeared to show more similarities with our assemblage. Specially, again,

concerning *G. inflata* as the main taxa. So now, *G. inflata* is the dominant taxa in the surface, intermediate depth (our assemblage) and the seabed sediment. However, this samples showed a high proportion of *G. ruber* (white). In turn, this species appears in a much reduced proportion in our samples (see next comment for the shell weight discussion).

**R1-C2: More specifically, it has been shown that planktonic foraminifera calcify in accordance with their habitat depth, with species dwelling at deeper levels producing heavier shells compared to those inhabiting surface waters (Zarkogiannis et al., 2022). For instance, *G. truncatulinoides* typically generates among the heaviest shells (see previous), as does *G. inflata* when compared to *G. ruber* and *G. bulloides* (Feldmeijer et., 2013). Consequently, certain current speeds may favor the settling of specific species specimens while others are dispersed elsewhere, potentially explaining the observed counts. In the central Mediterranean, *G. bulloides* is more prevalent in sediments (and thus resuspended sediment) than *G ruber*, contributing to a simultaneous peak in benthic foraminifera counts. Hydrodynamics may therefore account for discrepancies in cases where specific environmental drivers cannot be identified for certain species, leading to the invocation of other environmental controls in the text to explain the observed patterns. Indeed hydrodynamics in the area are particularly strong especially in spring (Gasparini et al., 2004), while sediment resuspension in the wider area is found to peak in spring (Grifoll et al., 2019) as well. Data from Copernicus also indicate increased flow speeds at 400m during spring of 2014.**

R1-R2: Authors agree that generally, the deep-dwelling species such as *G. truncatulinoides* and *G. inflata* are among the heaviest planktonic foraminifera species. Particularly compared to surface dwellers (Beer et al., 2010; Béjard et al., 2023). However, we find unlikely that differences in foraminifera shell weights could be responsible for the differences between the sediment trap and surface sediments. Firstly, and as mentioned in the previous comment, *G. inflata* is the most abundant species in both the surface BONGO nets, in our sediment trap, and in the seabed sediment (with [210]Pb chronology as a support, see Incarbona et al., (2019)). The abundances between these datasets are similar, highlighting a dominance of this species in the modern central Mediterranean Sea. Secondly, under the winnowing theory, *G. bulloides*, a lighter species very abundant in the sediment, should travel very far with strong currents and be very abundant in the surface waters , however, it is under represented in our sediment trap and even absent from surface BONGO nets collected during spring (Mallo et al., 2017). Thirdly, *G. ruber*, which is lighter than the deep-dwelling species, but heavier than some surface dwellers such as *G. bulloides* (Beer et al., 2010; Weinkauf et al., 2016). Interestingly, *G. bulloides* is almost twice more

abundant in our samples than *G. ruber*, under the winnowing theory, the opposite trend should be expected. Finally, it could be argued that *G. truncatulinoides*, as the heaviest species, should dominate the sediment trap samples, however, it is less abundant than *G. inflata*, which dominates the seabed sediment. In a winnowing theory, *G. truncatulinoides* should show a higher abundance in the seabed.

Finally, we also analyzed the data suggested by reviewer 1, from Copernicus, that states that the flow speed increased during spring 2014. The data we retrieved from 400m deep (mean Mediterranean sea water velocity) showed that the flow increase happened during February 2014. The flow speed for April 2014 is rather low. Both the monthly and daily data showed that the peak of current flow developed during winter. Which on the other hand also contradicts previous work such as Gasparini et al., (2004). However, as a conclusion of the previous two comments, we agree with reviewer 2 that, during spring, the current speed increase. Then we also agree the hydrodynamics might be playing a role and affect the distribution of taxa such as *G. bulloides* and *G. truncatulinoides*. Therefore, we added a new discussion chapter: **chapter 5.3**, named **"Influence of the hydrodynamic conditions on the planktonic foraminifera assemblage"**. It consists mainly of a discussion around the impact of hydrodynamics with all the bibliographic input provided by reviewer 2. In the end chapter we disclose the possibility of the winnowing impact during spring and overall, during higher current settings. In that regard, the **introduction** and **conclusion** have also been updated accordingly. **Lines 43-45 and 1058-1060**, respectively.

**R1-C3: MARGO site areas should be checked for sedimentation rates, as regions with high sedimentation will likely experience reduced hydrodynamics, facilitating the settling of lighter, surface-dwelling specimens. Additionally, for any inferences regarding ecosystem shifts in the Mediterranean, the sediment traps in other locations should be compared with nearby sedimentary material. Certainly, a pan-Mediterranean comparison should evaluate whether the data from the current study from the Sicily strait (area of high velocities) should be considered or disregarded.**

**R1-R3:** Authors agree with the suggestion that the sedimentation rates should be checked as they could provide useful information about the hydrodynamic context. Unfortunately, 9 of the retrieved core-tops appear as "unpublished", while the publications belonging to the remaining material do not provide the sedimentation rate (Thunell, 1978). Therefore, to provide a more complete picture of the seabed sediment assemblages, now the MARGO sites are not the only seabed sediment used, now the Incarbona et al., (2019) also appear in the manuscript. In the latter work, sites 342 and 407, located in the Sicily Channel slightly eastward of the C01 sediment trap, were analyzed and compared with water samples collected in the close vicinity of the C01 sediment trap. Additionally, and as mentioned earlier, the chronology is based on $^{210}$Pb.

In these samples, *G. inflata* also dominates the assemblages, followed by *G. ruber* and a similar proportion of *G. bulloides* as the one found in our samples. The MARGO samples are still discussed in **chapter 5.5**, but in much a reduced way.

Concerning the pan-mediterranean comparison, we did not include samples covering the whole Mediterranean basin because the novelty of our work is mainly the planktonic foraminifera data from the C01 sediment trap. In that regard, we wanted, firstly, to put this assemblage in perspective with other mooring lines (**chapter 5.4**) and then compare it with the seabed sediment in the central Mediterranean to possibly identify assemblages changes during recent times. There are various reasons why we limited our sediment comparison within the Sicily Channel. The distance between the C01 sediment trap and the seabed sediment is around 243km to the westernmost core-top (MARGO 3727) and 213km to the easternmost core-top (MARGO 3724), which in turn shows that the area covered is considerable. Also, we only included core-tops located in a 2.5 degres distance in order to only display potentially comparable seabed assemblages to the sediment trap. Finally, the concept of pan-mediterranean approach was not intended in our original version of the manuscript. Rather a "put into perspective" strategy.

However, as the comparison with the seabed sediment raised questions, we re-designed **chapter 5.5**. We now discuss the similarities and differences between the C01 sediment trap and the different seabed sediment datasets (MARGO and sites 342 and 407). We discuss the possibility of the retrieval device (i.e. core-top, box-core) impact on the sediment preservation. We now also discuss the similarities between our assemblages and the surface ones (Mallo et al., 2017). We also acknowledge the lack of sedimentation rates and the winnowing and the sediment resuspension as a possible explanation of the recent assemblages for the MARGO database, **lines 1012-1014**. The sum of the reasons we display within our comparison allow us to document a change in the planktonic foraminifera population during the Holocene, and. Although we propose the Incarbona et al., (2019) chronology, we cannot state with precision the exact timing of the latter. Note that the **introduction** and **conclusions** have been modified accordingly. **Lines 53-58 and 1066-1073.**

**R1-C4: Furthermore, in a future submission please change planktic to planktonic. The correct adjective form of plankton in Greek is planktonic. The adjectives of Greek nouns ending in -on get the suffix -ic in the end like plankton – planktonic, bion – bionic, lacon – laconic (preserved also in French words like Napoleon – Napoleonic). This is different to nouns ending in -os, which lose the ending -os to the previous consonant by replacing it with -ic, like bentos – benthic, cosmos – cosmic or chronos – chronic.**

**R1-R4:** Authors agree and understand the linguistic justification behind this comment. The term "planktic" has been replaced by "**planktonic**" in the whole manuscript. Note that not all the planktonic terms have been depicted in red, just a couple of examples.

**References**

Bárcena, M. A., Flores, J. A., Sierro, F. J., Pérez-Folgado, M., Fabres, J., Calafat, A., and Canals, M.: Planktonic response to main oceanographic changes in the Alboran Sea (Western Mediterranean) as documented in sediment traps and surface sediments, Marine Micropaleontology, 53, 423–445, https://doi.org/10.1016/j.marmicro.2004.09.009, 2004.

Beer, C. J., Schiebel, R., and Wilson, P. A.: Technical Note: On methodologies for determining the size-normalised weight of planktic foraminifera, Biogeosciences, 7, 2193–2198, https://doi.org/10.5194/bg-7-2193-2010, 2010.

Béjard, T. M., Rigual-Hernández, A. S., Flores, J. A., Tarruella, J. P., Durrieu De Madron, X., Cacho, I., Haghipour, N., Hunter, A., and Sierro, F. J.: Calcification response of planktic foraminifera to environmental change in the western Mediterranean Sea during the industrial era, Biogeosciences, 20, 1505–1528, https://doi.org/10.5194/bg-20-1505-2023, 2023.

Incarbona, A., Jonkers, L., Ferraro, S., Sprovieri, R., and Tranchida, G.: Sea Surface Temperatures and Paleoenvironmental Variability in the Central Mediterranean During Historical Times Reconstructed Using Planktonic Foraminifera, Paleoceanog and Paleoclimatol, 34, 394–408, https://doi.org/10.1029/2018PA003529, 2019.

Mallo, M., Ziveri, P., Mortyn, P. G., Schiebel, R., and Grelaud, M.: Low planktic foraminiferal diversity and abundance observed in a spring 2013 west–east Mediterranean Sea plankton tow transect, Biogeosciences, 14, 2245–2266, https://doi.org/10.5194/bg-14-2245-2017, 2017.

Rigual-Hernández, A. S., Sierro, F. J., Bárcena, M. A., Flores, J. A., and Heussner, S.: Seasonal and interannual changes of planktic foraminiferal fluxes in the Gulf of Lions (NW Mediterranean) and their implications for paleoceanographic studies: Two 12-year sediment trap records, Deep Sea Research Part I: Oceanographic Research Papers, 66, 26–40, https://doi.org/10.1016/j.dsr.2012.03.011, 2012.

Thunell, Robert C: Distribution of planktonic foraminifera in surface sediments of the Mediterranean Sea, https://doi.org/10.1594/PANGAEA.55624, 1978.

Weinkauf, M. F. G., Kunze, J. G., Waniek, J. J., and Kučera, M.: Seasonal Variation in Shell Calcification of Planktonic Foraminifera in the NE Atlantic Reveals Species-Specific Response to Temperature, Productivity, and Optimum Growth Conditions, PLoS ONE, 11, e0148363, https://doi.org/10.1371/journal.pone.0148363, 2016.

**RESPONSE TO REVIEWER 2:**

First of all, the authors would like to thank reviewer 1 for the input and all the ideas suggested. We think that the comments made helped to significantly improve the manuscript. In the manuscript, find the changes suggested by reviewer 1 depicted in red.

Here, to be clear and precise with our answers, we used **R#-C#** and bold notation for reviewer 1 comments, and our answers appear as **R#-C#.**

**R1-C1: While I acknowledge the authors' characterization of the Mediterranean as a 'miniature ocean' and recognize the significance of publishing foraminifera datasets from this relatively understudied sea, I maintain skepticism regarding the reliability of the results as an ecological signal. This skepticism arises primarily due to the presence of benthic specimens in the sediment trap, which allows for the interpretation of results influenced by hydrodynamics. In essence, the results represent a mixed signal of both ecology and hydrodynamics. For instance, the increased occurrence of deep-dwelling specimens may be attributed to the preferential settling of heavier specimens and the winnowing of lighter ones at the trap site. Similarly, the resemblance of the assemblage to that of the eastward core top sample could be a result of sediment winnowing from the predominantly westward flowing LIW water mass (200 m to 400 m) at the intermediate trap collection depth.**

**R1-R1:** Authors appreciate the point raised by reviewer 1. We agree with reviewer 1 in that the presence of benthic foraminifera raises the question of the role played by the hydrodynamics in the materials collected by the trap. There are several lines of evidence that strongly suggest that the foraminifera flux seasonality collected by the trap is mainly driven by changes in the production and export of foraminifera in the upper water column. Firstly, the seasonal cycle of planktonic production and export collected by the trap is consistent with many other settings within the Mediterranean and the world's oceans (Avnaim-Katav et al., 2020; Bárcena et al., 2004; Rigual-Hernández et al., 2012), displaying maximum values during winter and spring, thereby coinciding with the productive period. Secondly, the composition of the sinking foraminifer assemblages collected by the trap is in good agreement with the composition of the living foraminifera populations dwelling in the overlying water masses (Mallo et al., 2017). The latter used BONGO nets to analyze the planktonic foraminifera population during spring in a pan-mediterranean approach and documented that *G. inflata* was also the dominant species at the moment of sampling (during spring) alongside with *O. universa*. Please note, that this latter argument was not stated in the submitted version of the manuscript, but it is now in the updated version of the manuscript, **chapter 5.5, lines**

**919-921**. Lastly, it should be noted that although our data indicates that the fluxes collected by the trap are mainly a controlled by pelagic sedimentation, there is clear evidence of influence of resuspended materials into the trap as suggested by the presence of benthic foraminifera. However, we acknowledge that benthic foraminifera are registered at low numbers during the whole record (on average 3.3%), and this is the reason why background influence of resuspended materials was indicated in the manuscript (**lines 474-475** of the first version of the manuscript). In terms of contribution to the annualized foraminifera flux, benthic foraminifera only represented 1.1% of the total flux. Of those 1.1%, 80% was collected during the month of April 2014. However, we acknowledge that not all the arguments highlighted above were clearly explained in the first version of the manuscript and therefore, they will in the corrected version of the manuscript. Information about benthic foraminifera is now more complete at **chapters 4.1., 4.3 and 5.1., lines 358-360 , 402-404 and 514-525 respectively.** Note that more information about the benthics individuals was also requested by reviewer 1 and therefore appears

As an comparison, the Planier sediment trap, located in the axis of the Planier canyon in the Gulf of Lions and located 500 m above the seafloor registered 3.5% of benthic individuals with relative abundances ranging between 0.5-11% (unpublished data). In the Alboran Sea, the ALB5F sediment trap registered 2.9% of "benthic-neritic" (Bárcena et al., 2004). Our findings about benthic individuals are within the range of the previous work with which we compare our data. However, we acknowledge that the hydrodynamics are likely playing a role during the spring period.

Concerning the occurrence of deep dwellers, we acknowledge that the point raised by reviewer 1 is a possibility. In addition, as reviewer 2 suggested, we added a comparison with the work of Mallo et al., (2017). The latter used BONGO nets to analyze the planktonic foraminifera population during spring in a pan-mediterranean approach. The site located in the Sicilian Channel showed that, in surface waters, *G. inflata* dominated the assemblage, alongside with *O. universa*. Therefore, our most abundant species is in accordance with the most abundant from the surface record, which, in our opinion, rules out the resuspended origin of this taxa. The questions now transfers to the secondary taxa. *O. universa* seems to be particularly high alongside the Algerian coast (Azibeiro et al., 2023), however, the machanisms behind its distribution in the Mediterranean remain poorly constrained (Mallo et al., 2017).

Finally, concerning the LIW influence as a possible explanation for the resemblance of our assemblages with the eastern core-tops, the authors agree. Once again, reviewer 2 provided a useful input, which is to use the Incarbona et al., (2019) box-core samples to have a better constrained chronology. These samples were also retrieved in the Sicily Channel and appeared to show more similarities with our assemblage. Specially, again,

concerning *G. inflata* as the main taxa. So now, *G. inflata* is the dominant taxa in the surface, intermediate depth (our assemblage) and the seabed sediment. However, this samples showed a high proportion of *G. ruber* (white). In turn, this species appears in a much reduced proportion in our samples (see next comment for the shell weight discussion).

**R1-C2: More specifically, it has been shown that planktonic foraminifera calcify in accordance with their habitat depth, with species dwelling at deeper levels producing heavier shells compared to those inhabiting surface waters (Zarkogiannis et al., 2022). For instance, *G. truncatulinoides* typically generates among the heaviest shells (see previous), as does *G. inflata* when compared to *G. ruber* and *G. bulloides* (Feldmeijer et., 2013). Consequently, certain current speeds may favor the settling of specific species specimens while others are dispersed elsewhere, potentially explaining the observed counts. In the central Mediterranean, *G. bulloides* is more prevalent in sediments (and thus resuspended sediment) than *G ruber*, contributing to a simultaneous peak in benthic foraminifera counts. Hydrodynamics may therefore account for discrepancies in cases where specific environmental drivers cannot be identified for certain species, leading to the invocation of other environmental controls in the text to explain the observed patterns. Indeed hydrodynamics in the area are particularly strong especially in spring (Gasparini et al., 2004), while sediment resuspension in the wider area is found to peak in spring (Grifoll et al., 2019) as well. Data from Copernicus also indicate increased flow speeds at 400m during spring of 2014.**

R1-R2: Authors agree that generally, the deep-dwelling species such as *G. truncatulinoides* and *G. inflata* are among the heaviest planktonic foraminifera species. Particularly compared to surface dwellers (Beer et al., 2010; Béjard et al., 2023). However, we find unlikely that differences in foraminifera shell weights could be responsible for the differences between the sediment trap and surface sediments. Firstly, and as mentioned in the previous comment, *G. inflata* is the most abundant species in both the surface BONGO nets, in our sediment trap, and in the seabed sediment (with [210]Pb chronology as a support, see Incarbona et al., (2019)). The abundances between these datasets are similar, highlighting a dominance of this species in the modern central Mediterranean Sea. Secondly, under the winnowing theory, *G. bulloides*, a lighter species very abundant in the sediment, should travel very far with strong currents and be very abundant in the surface waters , however, it is under represented in our sediment trap and even absent from surface BONGO nets collected during spring (Mallo et al., 2017). Thirdly, *G. ruber*, which is lighter than the deep-dwelling species, but heavier than some surface dwellers such as *G. bulloides* (Beer et al., 2010; Weinkauf et al., 2016). Interestingly, *G. bulloides* is almost twice more

abundant in our samples than *G. ruber*, under the winnowing theory, the opposite trend should be expected. Finally, it could be argued that *G. truncatulinoides*, as the heaviest species, should dominate the sediment trap samples, however, it is less abundant than *G. inflata*, which dominates the seabed sediment. In a winnowing theory, *G. truncatulinoides* should show a higher abundance in the seabed.

Finally, we also analyzed the data suggested by reviewer 1, from Copernicus, that states that the flow speed increased during spring 2014. The data we retrieved from 400m deep (mean Mediterranean sea water velocity) showed that the flow increase happened during February 2014. The flow speed for April 2014 is rather low. Both the monthly and daily data showed that the peak of current flow developed during winter. Which on the other hand also contradicts previous work such as Gasparini et al., (2004). However, as a conclusion of the previous two comments, we agree with reviewer 2 that, during spring, the current speed increase. Then we also agree the hydrodynamics might be playing a role and affect the distribution of taxa such as *G. bulloides* and *G. truncatulinoides*. Therefore, we added a new discussion chapter: **chapter 5.3**, named **"Influence of the hydrodynamic conditions on the planktonic foraminifera assemblage"**. It consists mainly of a discussion around the impact of hydrodynamics with all the bibliographic input provided by reviewer 2. In the end chapter we disclose the possibility of the winnowing impact during spring and overall, during higher current settings. In that regard, the **introduction** and **conclusion** have also been updated accordingly. **Lines 43-45 and 1058-1060**, respectively.

**R1-C3: MARGO site areas should be checked for sedimentation rates, as regions with high sedimentation will likely experience reduced hydrodynamics, facilitating the settling of lighter, surface-dwelling specimens. Additionally, for any inferences regarding ecosystem shifts in the Mediterranean, the sediment traps in other locations should be compared with nearby sedimentary material. Certainly, a pan-Mediterranean comparison should evaluate whether the data from the current study from the Sicily strait (area of high velocities) should be considered or disregarded.**

**R1-R3:** Authors agree with the suggestion that the sedimentation rates should be checked as they could provide useful information about the hydrodynamic context. Unfortunately, 9 of the retrieved core-tops appear as "unpublished", while the publications belonging to the remaining material do not provide the sedimentation rate (Thunell, 1978). Therefore, to provide a more complete picture of the seabed sediment assemblages, now the MARGO sites are not the only seabed sediment used, now the Incarbona et al., (2019) also appear in the manuscript. In the latter work, sites 342 and 407, located in the Sicily Channel slightly eastward of the C01 sediment trap, were analyzed and compared with water samples collected in the close vicinity of the C01 sediment trap. Additionally, and as mentioned earlier, the chronology is based on $^{210}$Pb.

In these samples, *G. inflata* also dominates the assemblages, followed by *G. ruber* and a similar proportion of *G. bulloides* as the one found in our samples. The MARGO samples are still discussed in **chapter 5.5**, but in much a reduced way.

Concerning the pan-mediterranean comparison, we did not include samples covering the whole Mediterranean basin because the novelty of our work is mainly the planktonic foraminifera data from the C01 sediment trap. In that regard, we wanted, firstly, to put this assemblage in perspective with other mooring lines (**chapter 5.4**) and then compare it with the seabed sediment in the central Mediterranean to possibly identify assemblages changes during recent times. There are various reasons why we limited our sediment comparison within the Sicily Channel. The distance between the C01 sediment trap and the seabed sediment is around 243km to the westernmost core-top (MARGO 3727) and 213km to the easternmost core-top (MARGO 3724), which in turn shows that the area covered is considerable. Also, we only included core-tops located in a 2.5 degres distance in order to only display potentially comparable seabed assemblages to the sediment trap. Finally, the concept of pan-mediterranean approach was not intended in our original version of the manuscript. Rather a "put into perspective" strategy.

However, as the comparison with the seabed sediment raised questions, we re-designed **chapter 5.5**. We now discuss the similarities and differences between the C01 sediment trap and the different seabed sediment datasets (MARGO and sites 342 and 407). We discuss the possibility of the retrieval device (i.e. core-top, box-core) impact on the sediment preservation. We now also discuss the similarities between our assemblages and the surface ones (Mallo et al., 2017). We also acknowledge the lack of sedimentation rates and the winnowing and the sediment resuspension as a possible explanation of the recent assemblages for the MARGO database, **lines 1012-1014**. The sum of the reasons we display within our comparison allow us to document a change in the planktonic foraminifera population during the Holocene, and. Although we propose the Incarbona et al., (2019) chronology, we cannot state with precision the exact timing of the latter. Note that the **introduction** and **conclusions** have been modified accordingly. **Lines 53-58 and 1066-1073.**

**R1-C4: Furthermore, in a future submission please change planktic to planktonic. The correct adjective form of plankton in Greek is planktonic. The adjectives of Greek nouns ending in -on get the suffix -ic in the end like plankton – planktonic, bion – bionic, lacon – laconic (preserved also in French words like Napoleon – Napoleonic). This is different to nouns ending in -os, which lose the ending -os to the previous consonant by replacing it with -ic, like bentos – benthic, cosmos – cosmic or chronos – chronic.**

**R1-R4:** Authors agree and understand the linguistic justification behind this comment. The term "planktic" has been replaced by "**planktonic**" in the whole manuscript. Note that not all the planktonic terms have been depicted in red, just a couple of examples.

**References**

Bárcena, M. A., Flores, J. A., Sierro, F. J., Pérez-Folgado, M., Fabres, J., Calafat, A., and Canals, M.: Planktonic response to main oceanographic changes in the Alboran Sea (Western Mediterranean) as documented in sediment traps and surface sediments, Marine Micropaleontology, 53, 423–445, https://doi.org/10.1016/j.marmicro.2004.09.009, 2004.

Beer, C. J., Schiebel, R., and Wilson, P. A.: Technical Note: On methodologies for determining the size-normalised weight of planktic foraminifera, Biogeosciences, 7, 2193–2198, https://doi.org/10.5194/bg-7-2193-2010, 2010.

Béjard, T. M., Rigual-Hernández, A. S., Flores, J. A., Tarruella, J. P., Durrieu De Madron, X., Cacho, I., Haghipour, N., Hunter, A., and Sierro, F. J.: Calcification response of planktic foraminifera to environmental change in the western Mediterranean Sea during the industrial era, Biogeosciences, 20, 1505–1528, https://doi.org/10.5194/bg-20-1505-2023, 2023.

Incarbona, A., Jonkers, L., Ferraro, S., Sprovieri, R., and Tranchida, G.: Sea Surface Temperatures and Paleoenvironmental Variability in the Central Mediterranean During Historical Times Reconstructed Using Planktonic Foraminifera, Paleoceanog and Paleoclimatol, 34, 394–408, https://doi.org/10.1029/2018PA003529, 2019.

Mallo, M., Ziveri, P., Mortyn, P. G., Schiebel, R., and Grelaud, M.: Low planktic foraminiferal diversity and abundance observed in a spring 2013 west–east Mediterranean Sea plankton tow transect, Biogeosciences, 14, 2245–2266, https://doi.org/10.5194/bg-14-2245-2017, 2017.

Rigual-Hernández, A. S., Sierro, F. J., Bárcena, M. A., Flores, J. A., and Heussner, S.: Seasonal and interannual changes of planktic foraminiferal fluxes in the Gulf of Lions (NW Mediterranean) and their implications for paleoceanographic studies: Two 12-year sediment trap records, Deep Sea Research Part I: Oceanographic Research Papers, 66, 26–40, https://doi.org/10.1016/j.dsr.2012.03.011, 2012.

Thunell, Robert C: Distribution of planktonic foraminifera in surface sediments of the Mediterranean Sea, https://doi.org/10.1594/PANGAEA.55624, 1978.

Weinkauf, M. F. G., Kunze, J. G., Waniek, J. J., and Kučera, M.: Seasonal Variation in Shell Calcification of Planktonic Foraminifera in the NE Atlantic Reveals Species-Specific Response to Temperature, Productivity, and Optimum Growth Conditions, PLoS ONE, 11, e0148363, https://doi.org/10.1371/journal.pone.0148363, 2016.